


# Effect of chemical abrasion of zircon on SHRIMP U/Pb, δ18O, Trace element, and LA-ICPMS trace element and Lu-Hf isotopic analyses

Cate Kooymans[1], Charles W. Magee, Jr[1], Kathryn Waltenberg[1], Noreen J. Evans[2], Simon Bodorkos[1], Yuri Amelin[3, 4] Sandra L. Kamo[5], Trevor Ireland[3,6]

[1]Geoscience Australia, Symonston ACT, 2609, Australia
[2]Curtin University, Bentley WA 6102, Australia
[3]Australian National University, Canberra, ACT, 2600, Australia
[4]Korea Basic Science Institute
[5]University of Toronto, Department of Earth Sciences, Toronto, Ontario, Canada
[6]University of Queensland

*Correspondence to*: Charles W. Magee, Jr. (charles.magee@ga.gov.au)

**Abstract.** Chemical abrasion improves the U/Pb systematics of SHRIMP analyses of reference zircons, while leaving other
isotopic systems largely unchanged. SHRIMP $^{206}$Pb/$^{238}$U ages of chemically abraded reference materials TEMORA 2, 91500, QGNG, and OG1 are precise to within 0.25 to 0.4%, and are within uncertainty of chemically abraded TIMS reference ages, while SHRIMP $^{206}$Pb/$^{238}$U ages of untreated zircons are within uncertainty of TIMS ages of zircons which are untreated by chemical abrasion. Chemically abraded and untreated zircons appear to cross-calibrate within uncertainty using all but one possible permutations of reference materials, provided that the corresponding chemically abraded or untreated reference age
is used for the appropriate material. In the case of reference zircons QGNG and OG1, which are slightly discordant, the SHRIMP U-Pb ages of chemically abraded and untreated material differ beyond their respective 95% confidence intervals.
SHRIMP U/Pb analysis of chemically abraded zircons with multiple growth stages are more difficult to interpret. Treated igneous rims on zircons from the S-type Mount Painter Volcanics are much lower in common Pb than the rims on untreated zircons. However, the analyses of chemically abraded material show excess scatter. Chemical abrasion also changes the
relative abundance of the ages of zircon cores inherited from the sedimentary protolith, presumably due to some populations being more likely to survive the chemical abrasion process than others. We consider these results from inherited S-type zircon cores to be indicative of results for detrital zircons from unmelted sediments.
Trace element, δ18O, and εHf analyses were also performed on these zircons. None of these systems showed substantial changes as a result of chemical abrasion. The most discordant reference material, OG1, showed a loss of OH as a result of chemical
abrasion, presumably due to dissolution of hydrous metamict domains, or thermal dehydration during the annealing step of chemical abrasion. In no case did zircons gain fluorine due to exchange of lattice-bound substituted OH or other anions with fluorine during the HF partial dissolution phase of the chemical abrasion process. As the OG1, QGNG, and TEMORA 2 zircons are known to be compositionally inhomogenous in trace element composition, spot-to-spot differences dominated the trace element results. Even the 91500 megacrystic zircon exhibited substantial chip-to-chip variation. The LREE in chemically
abraded OG1 and TEMORA 2 were lower than in the untreated samples. Ti and phosphorus saturation ((Y+REE)/P) were generally unchanged in all samples.

## 1 Introduction

Three lines of development have driven the evolution of zircon U-Pb geochronology, from its inception (Holmes 1913) to the present-day. The first is improving precision and accuracy of Pb isotopic composition and U/Pb ratio measurements. The
second is developing sample treatments that allow extraction and analysis of domains in the zircons crystals that were closed



to migration of U and radiogenic Pb. The third is better understanding the sources and formation conditions of the zircons and their host rocks through analysis of zircon trace elemental, radiogenic and stable isotopic compositions.

In the last 30 years, significant progress has been achieved in all three areas. In zircon dating by isotope dilution thermal ionisation mass spectrometry (ID-TIMS), precision and accuracy were improved by development of highly efficient ion
emitters (Gerstenberger and Haase 1997) and by introduction of U and Pb double spikes made of high-purity synthetic isotopes (Chen and Wasserburg 1981, Todt et al. 1996), and combining these spikes for precise and accurate U-Pb and Pb-isotopic dating (Amelin and Davis 2006, Condon et al. 2015). Among the developments that help analysing closed U-Pb systems, chemical abrasion (Mattinson 2005; Mundil et al. 2004) is arguably the most important. Combining these developments, together with preparation of large quantities of carefully calibrated isotopic tracers in the Earthtime initiative (Condon et al.
2015; McLean et al. 2015), raised analytical precision and accuracy of U-Pb dating to a new level. At the same time, the measurement of multiple isotopic systems in individual zircon grains, in particular SIMS stable oxygen isotopes (Schuhmacher et al. 2004; Ickert et al. 2008) and the Lu-Hf system by both solution and laser ablation MC-ICPMS techniques (Amelin et al. 1999, Harrison et al. 2005, Hiess et al. 2009) has built geologic context around the U-Pb age, in terms of constraining the source material and petrogenesis of the melt from which the zircon crystallized.

Many, perhaps even most, of these developments are based on empirical findings enabled by researchers' intuition and refined through extensive experimenting, rather than being deduced from theory. As a result, these new techniques are often used without clear understanding of exactly how they work, and to what degree the various analytical techniques may interfere with one another. For example, we are not aware of any studies which determine whether chemical abrasion improves or complicates the analyses of stable oxygen isotope ratios by SIMS or Lu-Hf isotopic measurements by laser ICP-MS.

Chemical abrasion is applicable to analytical protocols other than TIMS U-Pb geochronology which involve analysis of dissolved zircon. Schoene et al. (2010) developed an approach to recover many elements (including Lu and Hf) from zircon dissolved for U-Pb geochemistry. This allows for their later use in elemental, Lu-Hf isotopic analysis as well as other isotopic systems such as $^{238}U/^{235}U$ (Hiess et al. 2012, Tissot et al. 2019) and Zr isotopes (Ibanez-Mejia and Tissot 2019) that emerge with ongoing analytical advances. However, dissolution of zircon in hydrofluoric acid makes recovering the zircon oxygen
and silicon isotopic compositions impossible. Also, phosphorus and titanium, the elements that are required for calculation of phosphorus saturation and for zircon crystallisation thermometry, were reported to be lost during zircon dissolution or chemical separation of Pb and U (Schoene et al. 2010). McKanna et al. (2023b) show that incompatible LREE are preferentially leached from zircon during the partial dissolution phase of chemical abrasion, but they only have trace element data on REE, Y, Nb, Hf, Ta, and Hf.

Several laser ICPMS studies (e.g. Crowley et al. 2014) have shown that chemical abrasion introduces matrix effects which cause apparent U-Pb age offsets of up to several percent. In contrast, SIMS studies have shown no appreciable effect for young zircons with low radiation damage (Watts et al. 2016), but results from zircons with more extensive radiation damage are consistent with the chemical abrasion process ameliorating Pb loss (Kryza et al. 2012, 2014). With the suggestion by Magee et al. (2023) that minor, cryptic Pb loss may be common in SIMS U-Pb dates from early Paleozoic and older zircons, SIMS
U-Pb dating of chemically abraded material could conceivably improve the precision and/or accuracy.

In-situ microbeam analysis of untreated zircon, both as unknowns and standards, involves an unresolved issue surrounding the standardization of U-Pb analysis. The CA-ID-TIMS revolution has bequeathed the in-situ geochronology community with a variety of newer, more precise reference ages for many of the zircons used as primary or secondary reference materials, many of which were initially characterised by ID-TIMS before the advent of chemical abrasion. For example, the laboratory
intercomparison study of Webb et al. (2023) shows that participating SIMS and LA-ICPMS labs use at least three different reference ages for a single reference zircon (91500). These ages are from both chemically abraded and untreated material.





Whether the CA TIMS age or the untreated TIMS age is more appropriate to use in microbeam techniques where matrix matching has traditionally been considered important has not been studied. As with the previous issues, a study of how chemical abrasion effects microbeam analyses of reference zircons would help inform this issue.

In this study, we take four well-known and widely used reference zircons spanning the timescale from Devonian to Paleoarchean, and compare how chemically abraded material performs against untreated material during U-Pb, oxygen isotope, and trace elemental analysis by SIMS, and U-Pb, Lu-Hf, and trace element analysis by laser ablation.

In this study of U-Pb systematics of zircon, we mainly focus on the changes in the $^{206}Pb/^{238}U$ ratio, because in nearly concordant zircons with a small degree of recent Pb loss and no ancient Pb loss, the $^{207}Pb/^{206}Pb$ ratio is almost constant, and any possible

variations are too small to be resolved by in-situ analysis. SHRIMP Pb isotopic fractionation was covered in Stern et al. (2009). Kositcin et al. (2011 ) show SHRIMP can produce $^{207}Pb/^{206}Pb$ ages with 2‰ precision, which correctly identify zircon grain subzones with ~1% difference in age (e.g. the Illogwa Shear Zone Mylonite in Kositcin et al. (2011)).

In addition to reference zircons, we analyse a population of zircons with distinct rims and cores from an S-type volcanic rock, to evaluate the effect of chemical abrasion on zircons with multiple growth domains. Multi-age zircon is often targeted with

microbeam techniques, and is relatively understudied by CA-ID-TIMS relative to simple igneous zircon.

**1.2 Samples**

Four reference zircons and one local igneous zircon were used in this study. The reference zircons were chosen to be old enough for decent counting statistics and different enough in age to span most of the timescale in which the Geoscience Australia SHRIMP laboratory works. They were TEMORA 2 (Black et al. 2004), 91500 (Wiedenbeck et al. 1995), QGNG

(Black et al. 2004), and OG1 (Stern et al. 2009). For this paper, the untreated versions of these four reference zircons are, respectively, T2U, 91U, QNU, and OGU, while the chemically abraded material is T2C, 91C, QNC, and OGC.

An S-type dacitic zircon from the Mount Painter Volcanics (Abell 1991) was also chosen. The Mount Painter Volcanics are part of the late Silurian volcano-sedimentary package that underlies the city of Canberra, Australia, and is part of the Lachlan Orogen. S-type igneous zircons in SE Australia often have igneous rims overgrowing older sedimentary cores, so the use of

this zircon allowed us to investigate the effect of chemical abrasion on zircon with multiple growth domains (Figure 1). This particular sample was chosen because the rims are fairly large (generally 25-75 μm) and easy to target with microbeam techniques. These volcanic zircon rims are also lower in U content than S-type granite rims, which often go metamict, and may not survive the chemical abrasion process. The untreated and chemically abraded material for this sample is referred to as MPU and MPC.

**2 Methods**

**2.1 Reference materials and values**

**2.1.1 SHRIMP U-Pb**

To evaluate the effect of chemical abrasion on SHRIMP U-Pb analyses, we need to consistently compare our SHRIMP results to ID-TIMS reference ages of both untreated and chemically abraded zircons. This is complicated by the fact that literature

ID-TIMS values for these reference zircons span 28 years, during which the methodology of ID-TIMS zircon geochronology has evolved.

Since the widespread adoption of chemical abrasion in the late 2000s, few U-Pb ID-TIMS ages of untreated reference zircons have been published, so reference ages for untreated zircons are generally from older papers than reference ages for chemically abraded zircons. Chemical abrasion improves precision of TIMS U-Pb analyses, but so do improvements to blanks, tracers,

mass spectrometers, and every other aspect of TIMS which occurred between the untreated reference value determinations and the chemically abraded reference value determinations.



One area where TIMS precision has improved over the last few decades is tracer uncertainty. Ideally we would use reference ages for all four reference zircons in both the chemically abraded and untreated state, determined using a single tracer. However, such a study has not yet been published. Instead, we chose reference ages calculated using the minimum number of different tracers for which we could find information.

There are three isotopic tracers used in our choice of reference values. They are (with samples used on):

- The Royal Ontario Museum (ROM) tracer (T2U, QNU, OGU, OGC); Black et al. (2003); Black et al. (2004); Stern et al. (2009);
- Earthtime 535 tracer (91U, QNC, T2C); Schoene et al. (2006), Schaltegger et al. (2021); and
- Earthtime 2535 tracer (91C, T2C); Horstwood et al. (2016), Schaltegger et al. (2021).

The two Earthtime tracers have identical $^{205}Pb/^{235}U$ ratios, which is the key ratio in determination of $^{206}Pb/^{238}U$ ages, and are considered to be the same for the purposes of this study, as Schaltegger et al. (2021) show no systematic difference in the age of TEMORA 2.

Using the published uncertainties for these tracers complicates intercomparison of untreated and chemically abraded material because the published uncertainty on the ROM tracer given in Black et al. (2003) is an order of magnitude higher than the Earthtime (McLean et al. 2015) tracer uncertainties. This causes the tracer uncertainty for the Black et al. (2003, 2004) and Stern et al. (2009) results to dominate the total uncertainty budget. Reducing this order of magnitude difference in tracer uncertainty makes intercomparison of CA and untreated results more straightforward.

The Earthtime project, in addition to creating the Earthtime tracer, also involved widely distributing several gravimetric solutions, which can be used to more precisely and accurately determine the isotopic ratios of tracer solutions. We are publishing ROM tracer results determined using all three of the gravimetric solutions described by Condon et al. (2015). These data were acquired after the Black et al. (2003, 2004) data, but before the Stern et al. (2009) data, making them relevant for the ROM lab at the time these reference values were determined. This allows us to reduce the reference value uncertainty for the (mostly untreated) samples analysed at the ROM to a level more commensurate with the Earthtime tracer, and well below SHRIMP analytical uncertainty.

For consistency, we recalculate all reference values using a single methodology. We take the weighted mean of the individual aliquots where available, and calculate the analytical uncertainty by multiplying the standard deviation of these results by Student's t. Where the probability of fit is less than 0.05, we also multiply by the square root of the MSWD. We also include four new aliquots for OGC, intended for the Stern et al. (2009) paper but not included in the final manuscript. For consistency, we apply the same calculation to the $^{206}Pb/^{238}U$ ratios from all of the reference analyses, generating the reference values and uncertainties given in Table 1. This difference in methodology accounts for the difference in these Table 1 values and their uncertainties compared to the headline numbers in the source papers.

Reference values derived from analyses using the same isotopic tracer do not need to propagate the tracer uncertainty when being compared to each other; similarly, the isotopic tracer uncertainty portion of the reference value uncertainty does not need to be propagated within SHRIMP sessions comparing two zircon reference values derived from the same tracer.

SHRIMP uncertainty propagation is described in Magee et al. (2023); In short, SHRIMP results in a single session can be compared to each other using just the sample analytical uncertainties (internal errors of Stern & Amelin 2003). However, when SHRIMP dates are compared to a TIMS reference value, the uncertainty of the SHRIMP reference zircon measurement for that session needs to be considered, as does the uncertainty on the SHRIMP reference zircon value. However, if a SHRIMP age is being compared to a TIMS reference age which used the same tracer as the reference zircon for the SHRIMP session, the tracer component of the reference zircon value should not be propagated. So, for example, a SHRIMP session using untreated TEMORA 2 (Black et al. 2004) as the reference zircon would not include the tracer portion of the reference zircon uncertainty when comparing the SHRIMP age for untreated QGNG to the reference TIMS age published in Black et al. (2003), as Black et al. (2003) and Black et al. (2004) both used the same tracer.



### 2.1.2 SHRIMP δ¹⁸O reference values

FC1 was used as the reference material with analyses distributed through the analytical session. A reference value of 5.6 ‰ was used (Avila et al. 2020).

### 2.1.3 SHRIMP Trace Element reference values

For negative ion multicollector work on SHRIMP SI, the Coble et al. (2018) value of 15ppm for 91500 was used to standardize fluorine contents of unknown zircons. No applicable OH values of any of the zircons studied here could be found, so $^{16}O^{1}H/^{18}O$ ratios are presented in raw form for data interpretation.

For positive ions, SHRIMP trace element concentrations were referenced to GZ7 (Nasdala et al. 2018) on our setup mount (GA5040). M127 (Nasdala et al. 2016) analyses from both the setup and experimental mounts, 91500 analyses from both mounts, and GZ8 (Nasdala et al. 2018) from the setup mount were used as secondary standards. For elements such as aluminium, which were not listed for GZ7 in Nasdala et al. (2018), the Coble et al. (2018) values for 91500 were used. The Szymanowski et al. (2018) isotope dilution value for titanium in GZ-7 was used for titanium concentrations to minimize the contribution of reference value uncertainty to the total uncertainty budget for titanium concentration determination.

### 2.1.5 LA-ICP-MS reference materials

The Laser ICP-MS split stream analyses used a setup mount containing 91500 (Wiedenbeck et al. 1995), Mud Tank (Woodhead and Hergt 2005), Plešovice (Slama et al. 2008), OG1 (Stern et al. 2009), GJ-1 (Jackson et al. 2004) zircons and NIST NBS 610 and 614 glass. Exact values and isotopic ratios used in the peak stripping process are given in the analytical methods below.

### 2.2 Sample preparation

Chemical abrasion of zircons was done at the Australian National University (ANU) in the manner of Huyskens et al. (2016). Aliquots of OG1, QGNG, 91500, Mount Painter, and TEMORA 2 zircon were chemically abraded by annealing at 900°C for 48 hours. Concentrated HF partially dissolved the annealed zircons at 190°C for 15 hours in a Parr bomb. After rinsing, the zircons returned to the Parr bomb for 15 hours at 190°C in HCl. A few hundred grains of both chemically abraded and unabraded zircons from each sample were then mounted in the centre 10mm x 10mm region of two 25mm epoxy disk mounts (mounts GA6363: reference zircons, and GA6364: Mount Painter Volcanics), produced according to the methods of DiBugnara (2016). After mount preparation and polishing, GA6363 and GA6364 were imaged in transmitted light, reflected light, and cathodoluminescence before being sputter-cleaned with argon and sputter-coated with 15nm of gold for surface conductivity during analysis.

### 2.3 Analytical Campaign

The analytical campaign involved making and photographing the mounts, performing U-Pb SHRIMP analyses in the Geoscience Australia geochronology laboratory, repolishing the spots off to prevent implanted ¹⁶O from compromising the next experiment, and analysing for δ¹⁸O, OH, and F on SHRIMP-SI at the Research School of Earth Sciences, Australian National University. After a preliminary analysis of the results, a subset of the previous spots was analysed for trace elements by SHRIMP at Geoscience Australia. Afterwards, mount GA6363 was taken to Curtin University for Laser Ablation Split Stream (LASS) trace element + Hf isotopic analysis. Due to the greater thickness of zircon needed for LASS analyses (tens of µm instead of 1µm), many of the laser analyses had to be relocated from where the SHRIMP spots were placed. Laser ablation analyses were not attempted on the Mount Painter Volcanic S-type zircons (mount GA6364) due to the possibility of rim-core drill-throughs complicating the interpretation.



During the data analysis, it was discovered that the initial SHRIMP trace element data for mount GA6363 were unsuitable,
due to a misplaced praseodymium peak. The laser holes were filled, the mount was repolished, and both mounts were
reanalysed for trace elements on the Geoscience Australia SHRIMP 2.

### 2.4 Analytical procedures

### 2.4.1 CA-ID-TIMS

The ID-TIMS analyses reported in Black et al., (2003; 2004) and Stern et al., (2009) were completed at the Jack Satterly
Geochronology Laboratory, Department of Earth Sciences at the University of Toronto, Canada, using the ROM tracer
solution. As part of the EARTHTIME Initiative in 2005, the ROM tracer was re-calibrated against 3 U-Pb gravimetric reference
solutions provided by the Initiative ('JMM', 'NIGL', 'MIT', results reported in Supplementary Table 1; c.f., Condon et al.,
2015). The aim at the time was to improve inter-comparability of dates reported by multiple laboratories by standardizing the
calibration of each U/Pb ratio for individually prepared tracer solutions.

Zircon grains which did not undergo chemical abrasion were mechanically air abraded (Krogh 1982). For the chemically
abraded OG1 TIMS results (Mattinson, 2005), zircons were thermally annealed at 1000°C for 48 hours and etched in 50%
hydrofluoric acid at 200°C for either 12 hours or 17 hours. Results for the chemically abraded OGC grains are reported in
Supplementary Table 1.

Zircon grains were rinsed in 7N $HNO_3$ at room temperature prior to dissolution. The ROM $^{205}Pb$-$^{235}U$ tracer was added to the
Krogh-type Teflon dissolution capsules during sample loading. The single zircon crystals were dissolved using ~0.10 ml of
concentrated HF acid and ~0.02 ml of 7N $HNO_3$ at 200° C for 4-5 days. Samples were dried to a precipitate and re-dissolved
in ~0.15 ml of 3N HCl overnight (Krogh, 1973). U and Pb were isolated from the bulk zircon solution using ~50 μl anion
exchange columns using HCl, dried in 0.05N phosphoric acid, deposited onto outgassed rhenium filaments with silica gel
(Gerstenberger and Haase 1997), and analyzed with a VG354 mass spectrometer using a single Daly detector in pulse counting
mode. Corrections to the $^{206}Pb$-$^{238}U$ ages for initial $^{230}Th$ disequilibrium in the zircon have been made assuming a Th/U ratio
in the magma of 4.2. All common Pb was assigned to procedural Pb blank. The dead time of the measuring system for Pb and
U was 16 and 14 ns, respectively. The mass discrimination correction for the Daly detector is constant at 0.05% per atomic
mass unit. Amplifier gains and Daly characteristics were monitored using the SRM 982 Pb standard. Thermal mass
discrimination corrections were 0.10% per atomic mass unit for both Pb and U. Decay constants are those of Jaffey et al.
(1971). All age errors quoted in the text, tables, and error ellipses in the concordia diagrams are given at the 95% confidence
interval. VG Sector software was used for data acquisition. In-house data reduction software in Visual Basic by D.W. Davis
was used. Plotting and age calculations were done using Isoplot 3.76 (Ludwig 2003).

### 2.4.2 SHRIMP U-Pb

The SIMS analyses were performed on the Geoscience Australia SHRIMP IIe. This is a single collector, duoplasmatron-only
SHRIMP installed in 2008 at Geoscience Australia by the manufacturer, Australian Scientific Instruments (ASI). The SHRIMP
has been upgraded over the years, specifically with a larger diameter post-ESA quadrupole lens for better refocusing of high
energy ions, and a piezoelectric stage. This stage, designed and built by ASI, uses three orthogonal SmarAct linear piezo
actuators to achieve sub-micron positional reproducibility in all directions. This reduces working distance changes and
secondary (QT1Y) steering variation between spots.

The SHRIMP extracted secondary ions at approximately 675V before accelerating them to 10kV and steering them into the
110 μm source slit of the Matsuda (1974) mass spectrometer. The collector slit was set to 100 μm, yielding a mass resolution
(M/ΔM) of approximately 5000 at the 1% peak height level. The energy window was left wide open to accept ions with an
energy spread of approximately -40 to +60 eV of forward energy, relative to the acceleration potential. After mass analysis,



ions were detected using an ETP electron multiplier. The retardation lens was not used. Electron multiplier dead time (25 ns) had previously been determined using Ti isotopic ratios in rutile. Analytical spots were programmed daily and run in approximately 23 hour batches.

For the standard zircons (mount GA6363, session 170123), after an initial concentration standard (zircon M127; Nasdala et al. 2016) was run, 42 spots were run on each of the eight zircon samples in a round robin fashion. A 100 µm Kohler aperture was

used, to produce an elliptical flat-bottomed sputter crater approximately 22 µm x 16 µm across and roughly 0.8µm deep. The primary beam monitor (PBM) measured a net sample current of 1.9 nA, which corresponds to a true primary beam current of 1.2 nA when analysing zircon. The acquisition table consisted of six scans through a 10 mass station run table: $^{90}Zr_2^{16}O$ (2 s), $^{204}Pb$ (20 s), Background ($^{204}Pb+0.05$ amu), (20 s), $^{206}Pb$ (15 s), $^{207}Pb$ (40 s), $^{208}Pb$ (5 s), $^{238}U$ (5 s), $^{232}Th^{16}O$ (2 s), $^{238}U^{16}O$ (2 s), $^{238}U^{16}O_2$ (2 s).

For the S-type zircons (mount GA6364, session 170124), 36 rims from both the CA and untreated aliquots of Mount Painter Volcanics zircon were run. This was followed by approximately 70 core analyses on each sample, in the manner of a sedimentary detrital zircon study. Untreated TEMORA 2 (Black et al. 2004) zircon was used as the primary reference zircon, with untreated 91500 and untreated OG1 zircon run as the secondary reference zircon and $^{207}Pb/^{206}Pb$ reference zircon, respectively. The run table and other analytical conditions were unchanged from the previous session. A quick follow-up

session (210046) was run using the same settings on those chemically abraded Mount Painter Volcanics rims which initially had anomalously young or old ages.

SHRIMP U-Pb data were processed using Squid 2.5 (Ludwig 2009). This software deadtime-corrects, background subtracts, and normalizes the data to the secondary beam monitor (SBM) to remove the effects of changes in the secondary beam intensity, before using Dodson (1978) interpolation to calculate isotopic ratios. The $^{204}Pb$ isotope was used for common Pb

correction of both the reference zircon and the unknowns, as $^{204}Pb$ overcounts were within uncertainty of zero for all sessions. While routine geochronology at Geoscience Australia is done using a ln(Pb/U) vs ln(UO/U) calibration with slope 2 (Claoue-Long et al. 1995), a detailed examination of these data showed that only three of the eight calibration slopes were within uncertainty of that value (Figure 2). Furthermore, the slope for zircon 91500 was shallower than the slopes of the other reference zircons. Chemical abrasion did not seem to have any substantial effect on calibration slope (Figure 2).

In this experiment, we choose to apply a calibration slope of 1.8. This is the best fit for the three non-megacryst samples, and it is the value which the Geological Survey of Canada (GSC) uses for their SHRIMP data reduction (Rayner, pers. comm.). Whether a systematic error component is required for 91500 data for having a shallow slope will be addressed in the discussion section.

Stern and Amelin (2003) determined that using the GSC SHRIMP, the spot-to-spot variability in calibrated Pb/U ratios in 610

glass was on the order of 1%. As the glass is homogenous for these elements at this level, this "spot-to-spot error" was accounted for in most subsequent SHRIMP data reduction procedures, with minimum values somewhere between 1% or 0.5% often used in data reduction. As the purpose of this study is to see if chemical abrasion, automated analysis and piezoelectric positioning can improve this number, for this study we start with a default spot-to-spot error of zero, and assign spot-to-spot uncertainty expansion only if the probability of fit for the calibration line in the primary reference material is less than 0.05.

**2.4.3 SHRIMP δ¹⁸O**

The analytical procedures for SHRIMP SI oxygen isotope analysis closely follows those employed by Avila et al. (2020). A ca. 5 nA $Cs^+$ primary ion beam is focused to a 25x20 µm spot. Charging is neutralised through focusing of a 2.2 kV electron beam on to the sputter area. Oxygen isotopes were measured in multiple collection mode with $^{16}O$ and $^{18}O$ measured across $10^{11}$ Ω resistors. Data were reduced with the ANU data reduction program POXI.



### 2.4.4 SHRIMP OH and F

While the OH peak with a nominal mass of 17 amu was measured during the $\delta^{18}O$ measurements, an additional experiment was run in which the $^{16}O^1H^-(17)$, $^{18}O^-$, and $^{19}F^-$ ions were simultaneously collected and measured. This is because zircon can contain structural OH in the lattice (Trail et al. 2011), and we wished to determine whether the HF dissolution step might also result in F for OH substitution in the structurally sound zircon matrix during chemical abrasion. The analytical procedure is

similar to that of Beyer et al. (2016), where OH in the low mass faraday cup and fluorine in the high mass cup are both ratioed to $^{18}O$ in the centre cup. Fluorine concentration was normalized using a 91500 concentration of 15 ppm (Coble et al. 2018).

### 2.4.5 SHRIMP trace elements

SHRIMP trace element analyses were done on the SHRIMP IIe single collector instrument at Geoscience Australia following the LASS analyses and epoxy filling of the laser holes. The primary beam was a ~1.9 nA (net current; as measured by the

Primary Beam Monitor- true current approximated at 1.2 nA) beam of $O_2^-$ ion focused into an 16 µm x 22 µm spot through the use of a 100 µm Köhler aperture. The method used was similar to Beyer et al. (2020), but with a few changes in mass stations.

Energy filtering was used to exclude low energy secondary ions. The low energy shutter was inserted 5.5 mm, sufficient to reduce the $^{238}U$ peak on metamict zircon by 90%. In order to optimize the extraction of high energy ions from the sample, and

transmission from the sample through the source slit of the mass spectrometer, the extraction voltage was dropped from 675 V to 625 V. The total acceleration remained 10 kV, with the difference in voltage accelerating the ions between the extraction plate and the acceleration cone.

The magnet cycled through a run table containing the following masses: $^{16}O^+$, $^{19}F^+$, $^{27}Al^+$, $^{30}Si^+$, $^{31}P^+$, $^{44}Ca^+$, $^{28}Si^{16}O^+$, $^{49}Ti^+$, $^{56}Fe^+$, $^{89}Y^+$, $^{90}Zr^{28}Si^{16}O^+$, $^{139}La^+$, $^{140}Ce^+$, $^{143}Nd^+$, $^{146}Nd^+$, $^{147}Sm^+$, $^{149}Sm^+$, $^{153}Eu^+$, $^{155}Gd^+$, $^{157}Gd^+$, $^{159}Tb^+$, $^{161}Dy^+$, $^{163}Dy^+$, $^{165}Ho^+$,

$^{166}Er^+$, $^{167}Er^+$, $^{169}Tm^+$, $^{171}Yb^+$, $^{172}Yb^+$, $^{175}Lu^+$, $^{179}Hf^+$, $^{180}Hf^+$, $^{232}Th^+$, $^{238}U^+$.

The elements F, Al, P, Ca, and Fe were standardized using the reference zircon 91500. All other elements were standardized using the G7 reference zircon. Reference zircons M127 and G8 (Nasdala et al. 2016, 2018) were used as secondary reference materials. Uncertainties for each spot analyses were calculated by adding in quadrature the uncertainty from the analytical spot to the uncertainty of the weighted mean of the reference zircon and the uncertainty on the reference value for that zircon. As

all the zircons studied aside from 91500 are zoned, and contain substantial trace element variations, the median value was reported for each sample.

Orthogonal polynomial shape coefficients (O'Neill 2016) as adopted for zircon (Burnham 2020) were calculated using the Anenburg and Williams (2022) web tool.

### 2.4.6 Laser ablation split stream ICP Hf isotopic and trace elemental analyses

Hf isotopes and U-Pb ages in zircon were simultaneously measured by laser ablation split stream at the Geohistory facility in the John de Laeter Centre, Curtin University, Western Australia. Zircon crystals mounted in 25mm epoxy rounds were ablated using a Resonetics resolution M-50A incorporating a Compex 102 excimer laser, coupled to a Nu Plasma II multi-collector inductively coupled plasma mass spectrometer (MC-ICPMS) for Hf isotope determination and an Agilent 7700 quadrupole inductively coupled plasma mass spectrometer (Q-ICP-MS) for age and trace element determination. Following two cleaning

pulses and a 40s period of background analysis, samples were spot ablated for 40s at a 10Hz repetition rate using a 50µm diameter beam and laser energy at the sample surface of 2.2Jcm$^{-2}$. An additional 15s of baseline was collected after ablation. The sample cell was flushed with ultrahigh purity He (320 mL min$^{-1}$) and $N_2$ (1.2 mL min$^{-1}$) and high purity Ar was employed as the plasma carrier gas, split to each mass spectrometer.

For Hf isotope analysis, all isotopes ($^{180}Hf$, $^{179}Hf$, $^{178}Hf$, $^{177}Hf$, $^{176}Hf$, $^{175}Lu$, $^{174}Hf$, $^{173}Yb$, $^{172}Yb$ and $^{171}Yb$) were counted on the

Faraday collector array. Time resolved data was baseline subtracted and reduced using Iolite (DRS after Woodhead et al.,





2004), where $^{176}$Yb and $^{176}$Lu were removed from the mass 176 signal using $^{176}$Yb/$^{173}$Yb = 0.7962 (Chu et al., 2002) and $^{176}$Lu/$^{175}$Lu = 0.02655 (Chu et al., 2002) with an exponential law mass bias correction assuming $^{172}$Yb/$^{173}$Yb = 1.35274 (Chu et al., 2002). An effective $^{176}$Yb/$^{173}$Yb correction factor was determined for each session by iteratively adjusting the $^{176}$Yb/$^{173}$Yb ratio until standard corrected ratios on secondary zircon reference materials with varying Yb content yielded values within the

recommended range. No correlation was apparent between the abundance of interfering isotopes (Yb or Lu) and corrected $^{176}$Hf/$^{177}$Hf ratios. The interference corrected $^{176}$Hf/$^{177}$Hf was normalized to $^{179}$Hf/$^{177}$Hf = 0.7325 (Patchett and Tatsumoto, 1980) for mass bias correction. Zircons from the Mud Tank Carbonatite locality were analysed together with the samples in each session to monitor the accuracy of the results. Twenty analyses of Mud Tank zircon yielded a $^{176}$Hf/$^{177}$Hf value of 0.282507 ± 20 (MSWD = 0.8) identical within uncertainty to the recommended value (0.282505 ± 44; Woodhead and Hergt,

2005). OG1 and Plešovice zircons were run to verify the method with weighted average $^{176}$Hf/$^{177}$Hf values (OG1 = 0.280607±0.000027, MSWD = 0.87, n =15; Plešovice 0.282466±0.000023, MSWD = 1.2, n = 10) determined within uncertainty of their accepted values (OG1 = 0.280560 ± 20, Kemp et al., 2017; Plešovice = 0.282482 ± 0.000013, Slama et al., 2008). In addition, the corrected $^{180}$Hf/$^{177}$Hf ratio was calculated to monitor the accuracy of the mass bias correction and yielded an average value of 1.886868 ± 17 (MSWD = 1.3), which is within the range of values reported by Thirlwall and Anczkiewicz

(2004). Calculation of εHf values employed the decay constant of Scherer et al. (2001) and the Chondritic Uniform Reservior (CHUR) values of Bouvier et al. (2008).

For the Q-ICP-MS analysis, the following elements were monitored for 0.01 s each, unless otherwise noted: $^{28}$Si, $^{31}$P, $^{44}$Ca, $^{49}$Ti (0.05 s dwell), $^{89}$Y, $^{90}$Zr, $^{139}$La, $^{140}$Ce, $^{141}$Pr, $^{146}$Nd, $^{147}$Sm, $^{153}$Eu, $^{157}$Gd, $^{163}$Dy, $^{166}$Er, $^{172}$Yb, $^{175}$Lu, $^{201}$Hg, $^{204}$Pb, $^{206}$Pb, $^{207}$Pb,

$^{208}$Pb (0.1 s dwell time on all Pb isotopes), $^{232}$Th (0.025 s dwell time), and $^{238}$U (0.025 s dwell time). International glass standard NIST 610 and reference zircon GJ-1 were used as primary standards to calculate elemental concentrations and to correct for instrument drift (using $^{29}$Si and $^{90}$Zr as the internal standard elements, respectively and assuming 14.26% Si and 43.14% Zr in the zircon unknowns). NIST 610 was the primary reference material for P, Ca, Zr, Pb, Th and U determination, while GJ-1 was the primary reference material for Ti, Y, La, Ce, Pr, Nd, Sm, Eu, Gd, Tb, Dy, Er, Yb and Lu. NIST 614 was treated as a

secondary standard for trace element determination with most elements reproducing within 5% of the recommended value.

The primary dating reference materials used in this study were 91500 (1063.55±0.4 Ma; Schoene et al., 2006) and OG1 (3465.4±0.6 $^{207}$Pb/$^{206}$Pb age for isotopic fractionation monitoring; Stern et al., 2009) with Plesovice (337.13±0.37 Ma; Sláma et al., 2008) and GJ-1 (608.53±0.37; Jackson et al., 2004) analysed as secondary $^{206}$Pb/$^{238}$U age standards. $^{206}$Pb/$^{238}$U ages and

$^{207}$Pb/$^{206}$Pb calculated for zircon age standards, treated as unknowns, were found to be within 3% of the accepted value. The time-resolved mass spectra were reduced using the U_Pb_Geochronology4 data reduction scheme in Iolite 3.5 (Paton et al, 2011 and references therein).

The laser spots were run in a different order to the SHRIMP spots, and grain identities were not preserved. A table matching up laser and SHRIMP grain numbers for mount GA6363 (reference zircons) can be found in supplementary table S2.

Additionally, the supplementary sample maps, which show all spot analyses are in supplementary Figures S1 (reference zircons) and S2 (Mount Painter Volcanics).

## 3 Results

### 3.1 ROM calibration and reference zircon age recalculation.

Seven aliquots of the gravimetric reference solutions described by Condon et al. (2015) were run at the Royal Ontario Museum.

These were three replicates each of the JMM and RP solutions, and one of the MIT solution. The results are given in supplementary table S1. The weighted mean $^{235}$U/$^{205}$Pb ratio of the ROM tracer solution was 106.569 ± 0.1 (2σ), with a MSWD



of 0.177 and a very high probability of fit of 0.983. This is well within the previous estimate of 106.54 ± 0.28 (2σ) given in Black et al. (2003). Following the advice of Condon et al. (2015), the central value, and therefore the reference $^{206}$Pb/$^{238}$U ratio of the reference zircons, was not changed.

McLean et al (2015) show that the contribution of the tracer uncertainty is smaller than the total analytical uncertainty due to error correlations in the calculations of the tracer solution composition. As we used the same gravimetric solutions as McLean et al. (2015) and the ROM tracer has a similar $^{235}$U/$^{205}$Pb ratio of ~100, we scale the tracer uncertainty contribution by 0.53, in a conservative approximation of the scaling of McLean et al. (2015). This gives us a tracer uncertainty contribution of approximately 0.05% to the systematic uncertainty.

Using this new uncertainty value, we recalculated the reference values for all reference zircons, which are given in Table 1.

**3.2 SHRIMP U-Th-Pb results of reference zircons**

SHRIMP session 170123 generally ran without incident; only a single analysis (T2C.39.1) had to be discarded due to instrumental instability producing a nonsensical downhole fractionation pattern. $^{204}$Pb overcounts were within error of zero, and the $^{207}$Pb/$^{206}$Pb ratio (Tables S3, S4) for both untreated and chemically abraded OG1 were within error of their respective

reference values, indicating no detectable mass-based isotopic fractionation. Individual spot data reduced using the T2U as the primary reference material and Black et al. (2004) as reference value is presented in Supplementary Table S3. Individual spot data reduced using T2C as the primary reference zircon and the reference value of Schaltegger et al. (2021) are presented in Supplementary Table S4. Measured weighted mean ages of all samples, relative to either T2U or T2C, are shown in Table 2. Weighted means of the spot averages using T2C as the primary reference material, and their comparison to the TIMS reference

values are shown in Figure 3, panels 1-7. Calibration slope probabilities of fit were better than 0.05 for all chemically abraded samples, and for the T2U and 91U zircons, indicating that no excess spot-to-spot error was required in the reduction of this data set for those reference zircons known to reliably exhibit closed system U-Pb behaviour.

In all cases, chemical abrasion reduced the 95% confidence envelope of the mean, reduced the MSWD, and increased the probability of fit for the weighted mean for unknowns, regardless of which zircon was chosen as the primary reference. On

average, the chemically abraded grains have slightly larger intraspot uncertainty, but this is consistent with worse counting statistics from lower average uranium contents, which we attribute to survivor bias in the chemical abrasion process. The lower MSWD for the chemically abraded samples is not simply a result of larger single spot uncertainty; the central values are also less dispersed. For the chemically abraded samples, the analytical 95% confidence interval on the means was on the order of ± two permille. All chemically abraded ages were within uncertainty of their reference TIMS ages, when either T2U or T2C

is used as the primary reference material.

The ages for 91U and 91C were within uncertainty of each other, as were T2U and T2C. OGU and QNU, however were younger, and had a dispersed population and a high MSWD, compared to their chemically abraded counterparts. The population mean, however, had an age consistent with the TIMS ages of untreated, not chemically abraded zircon (Stern et al. 2009 and Black et al. 2003), and not with the chemically abraded ages for those samples (Table 2, Figure 3, Figure 4).

Of course, any of these zircons can be used as the reference zircon instead of TEMORA 2. The only pairing of reference zircon and unknown which does not result in the samples being within error of their reference values is the pair of 91U and OGC (or vice versa), which report an offset on the unknown relative to the reference value of approximately 0.45% (younger if 91U is the reference and OGC is the unknown, older if the reverse), which exceeds the precision of these measurements. All other reference-unknown combinations result in ages within uncertainty of the reference ages.

Jeon and Whitehouse (2014) showed that for their 1280 SIMS instrument, calibration equations which used the UO$_2$ peak instead of the $^{238}$U peak were more precise. We checked all of the potential calibration equations presented in Jeon and Whitehouse (2014) to determine if further increases in precision could be achieved. In contrast to their results, we find that those calibration equations which use $^{270}$(UO$_2$) instead of (or in addition to) $^{238}$U offer no improvement relative to the



$^{206}Pb/^{238}U$ vs $^{254}UO/^{238}U$ calibration of Claouè-Long et al. (1995). A summary of all 8 calibration equations is shown in
Table 3. Note that as the calibration variation experiment was performed using a floating calibration slope, there are slight
differences between these data and the fixed slope results reported above and in Table 2.

### 3.3 SHRIMP U/Pb of Mount Painter Volcanics zircons

SHRIMP session 170124 did not run as smoothly as session 170123. The untreated TEMORA 2 (Black et al. 2004) primary
reference material on this mount had an MSWD of 1.71, a probability of fit of 0.0001, and a spot-to-spot error of 0.61%. The
standard error on the 76 TEMORA 2 grain calibration was 0.11%. The spot level results for both inherited cores and igneous
rims are reported in table S5.
The weighted mean geochronological results are listed in Table 4. One chip of the untreated secondary reference zircon of
91500 gave two analyses suggesting Pb loss, so these spots were excluded from the mean. The other 16 analyses yielded a
$^{206}Pb/^{238}U$ age of 1060.1 ± 7.0 Ma. Untreated reference zircon OG1 gave a $^{206}Pb/^{238}U$ age of 3436.2 ± 15.5 Ma. Both of these
ages are within uncertainty of the reference values in table 1 for untreated zircon. The OG1 $^{206}Pb/^{238}U$ age is younger than the
$^{207}Pb/^{206}Pb$ age of this sample, and is consistent with the OG1 age of untreated OG1 ages given in part 3.1 of this paper (Tables
2a, 2b, Figure 3), as well as with several OG1 U/Pb ages reported from this lab over the past decade summarized by Magee et
al. (2023).

### 3.3.1 Igneous age of Mount Painter Volcanics zircons

Seven of the Mount Painter untreated zircon rims have over 1% common $^{206}Pb$, the highest of which is 16%. Although common
Pb corrections pull these analyses back into the same population as the low Pb analyses, we still exclude them from the
weighted mean. The remaining 29 untreated Mount Painter rims give a $^{206}Pb/^{238}U$ age of 429.7 Ma ±1.3/1.7 Ma (internal /
external). All 29 analyses define a single population, with a MSWD of 1.12 and a probability of fit of 0.30.
The chemically abraded rims are devoid of high common Pb analyses, with common Pb content less than 0.2% for all spots.
Despite this, the results are somewhat more complicated, as the 36 analyses are dispersed, even after including the 0.61% spot-
to-spot error. One of these, spot MPC.21.1, seems to have clipped the edge of a core based on post-analysis CL images, and
this is excluded from further consideration. The remaining 35 analyses have a weighted mean age of 431.8 Ma, ±1.7/2.0 with
an MSWD of 1.97 and a probability of fit of 0.0006. A probability of fit greater than 0.05 can be achieved by discarding two
additional outlier grains, one high and one low, to give an age of 431.6 ±1.2/1.6 Ma. The non-grouping chemically abraded
samples from the Mount Painter Volcanics- both cores and rims- were reanalysed in session 210046 to see if the difference in
age was compositional or analytical (see discussion). Data for session 210046 is in supplementary Table S6.
In addition to targeting the rims of these zircons for an igneous age, we also dated 78 cores from both the chemically abraded
and untreated aliquots of Mount Painter zircons. In both samples the cores yielded a range of ages, but in each case the youngest
core population was within uncertainty of the rim age.
In the untreated samples, a population of the six youngest cores gave a pooled age of 430.1 ± 4.1/4.2 Ma. The MSWD was
1.31, giving a probability of fit of 0.25.
For the chemically abraded samples, there were 18 young cores, which yielded an age of 431.3 ±1.8/2.1 Ma. The MSWD was
1.31 with a probability of fit of 0.17.
As these populations are indistinguishable from the rim ages, we can combine the youngest cores and the rims to report pooled
ages. These give ages of 429.8 ± 1.2/1.6 Ma for the MPU core+rim, and 431.3 ± 1.0/1.5 Ma for the MPC core+rim zircons.

### 3.3.2 Mount Painter Volcanics inherited ages

Most of the cores in both Mount Painter samples were older than the igneous age. None were younger. The probability density
diagrams of the cores younger than 1200 Ma are given in Figure 5. There are scattered individual Paleoproterozoic grains in



both samples, but they do not form discrete populations in either sample. In both MPC and MPU, the youngest population of
cores was within uncertainty of the rim age. However, in the chemically abraded sample, the proportion of these cores was
three times larger than in the untreated zircon population. Because these cores are indistinguishable in age from the rims for
both the untreated and chemically abraded samples, a combined age for both is presented in Table 4, which yields slightly
more precise ages than the cores alone due to the larger sample size. It is worth noting, however, that the ~430Ma cores have
a higher median Th/U ratio (0.37) than the ~430 Ma rims (Th/U=0.15), and thus may represent an earlier (based on core-rim
geometry, not U-Pb age) magma chamber process than the rims.

It is not only the ~430 Ma cores which change their relative abundance with chemical abrasion. The 550-610 Ma population
of MPC is only about half as large as in MPU (figure 5).

### 3.4 SHRIMP δ¹⁸O results

As the SHRIMP SI can hold up to three round mounts, both GA6363 (standards) and GA6364 (Mount Painter) were loaded
and run as a single session. 20-25 spots on each reference zircon were run, as well as 20 spots of the rims of MPC and 25 spots
on the MPU rims. 35 spots were put on MPU cores, while 30 spots were put on MPC cores. The $\delta^{18}O$ results are in Table 5,
with complete spot by spot data in supplementary table S7. Figure 6 shows plots of $\delta^{18}O$ of the samples.

### 3.5 SHRIMP trace element results

After the $\delta^{18}O$ session reported above, the SHRIMP SI magnet was incremented by 1 amu and the cups were repositioned to
measure $^{16}O^1H^-$, $^{18}O^-$, and $^{19}F^-$ to examine the OH-F systematics in a single analytical volume. The OH-$^{18}$O-F spot by spot
results are in supplementary table S8, and the OH/$^{18}$O ratios of the samples are plotted in Figure 7.

The results for all REE, Hf, Th, U, Y, Ti, P, Al, Ca, Fe and F measured by SHRIMP IIe as positive ions are given in Table 6.
Full spot-by-spot data are in supplementary table S9 for the reference zircons on mount GA6363 and Mount Painter Volcanics
zircons on mount GA6364. All results are µg/g. Orthogonal polynomial coefficients (λ; O'Neill 2016) are presented along
with titanium content-based rutile equilibrium temperatures (t(Ti)) calculated using Ferry and Watson (2007). Phosphorus
saturation (Burnham and Berry 2017) is also shown. REE patterns for the reference zircons are shown in Figure 8, while the
REE patterns for the Mount Painter Volcanics zircon rims are shown in Figure 9.

### 3.6 LASS U-Pb, trace element and Lu-Hf results

### 3.6.1. LASS U-Pb results

The laser ICPMS U-Pb geochronology results are presented in Table 7. Total external uncertainties within the $^{206}Pb/^{238}U$
system were generally about 0.5 to 1% for all untreated samples.  Half of samples were within the stated uncertainty of their
reference values. The laser results for T2C (410.7 ± 1.5) and 91U (1047.9 ± 8.9) were too young. The laser results for QNU
(1859.6 ± 10.2) and OGU (3473.3 ± 16.4) were too old for the untreated reference values, but within uncertainty of the
chemically abraded reference values. With the exception of 91500, the chemically abraded samples were all younger than the
untreated grains. However, this difference was only statistically significant for QGNG. This is the opposite sense to that
expected from a physical process such as the removal of damaged discordant zircon, and is probably the instrumental effect
documented by Crowley et al. (2014) and not a physical change in the sample. Chemically abraded 91500, which had a much
smaller grainsize than untreated 91500 and suffered more burn-through analyses as a result, appears to have been more affected
by common Pb. This may be a surface or epoxy contaminant entrained into the gas flow to the torch when the laser burned
through the back or sides of the grain. Spot by spot laser U-Pb data are presented in supplementary table S10.



### 3.6.2. LASS trace element results

Trace elements were analysed in the same quad ICP mass cycles as the U-Pb isotopes. Due to the dwell time required for Pb isotopes, the LREE aside from cerium were often below detection limits. Many samples had lanthanum and praseodymium below detection limit (BLD), and in 91500 most of the LREE were BLD. As a result, orthogonal polynomial coefficients
(O'Neill 2016) were not calculated. Spot by spot results are listed in supplementary table S11.

### 3.6.3. LASS Lu-Hf results

The laser ablation split stream sent half the ablated material from the U-Pb and trace element analyses described above to a multicollector ICPMS for Yb-Lu-Hf isotopic analysis. The number of samples is therefore the same as mentioned above. The multicollector-based Yb/Hf ratio for each spot is consistent with the trace element data. Table 8 and Figure 10 show the
weighted mean initial Hf isotopic compositions, as ratios and as εHf(t). As these geochronology reference zircons have variable Lu-Hf ratios, the mean measured Hf isotopic values are of course more scattered due to variable amounts of ingrowth, particularly for the older zircons (QGNG and OG1). The spot-to-spot data, including the measured $^{176}$Hf/$^{177}$Hf and $^{176}$Lu/$^{177}$Hf are in supplementary Table S12.

### 4 Discussion

### 4.1 ROM tracer recalibration and reference values

The recalibration of the ROM tracer reduces the systematic uncertainty by a factor of five relative to the values published in Black et al. (2003, 2004). This in turn reduces the uncertainty in the reference ages by 140-290%, depending on the reference zircon (noting that only Black et al. (2004) explicitly report an uncertainty including the tracer; Black et al. (2003) and Stern et al. (2009) leave that calculation as an exercise for the reader). As a result, the tracer uncertainty is now much smaller than
any of the other uncertainty components from these U-Pb SHRIMP results, allowing us to compare the results without the complication of an order-of-magnitude tracer uncertainty difference. We recommend the reference values in Table 1 be used for all listed non-chemically abraded reference zircons used to standardize in-situ analyses for this reason.

### 4.2 SHRIMP U-Pb analyses of reference zircons

SHRIMP U-Pb analyses of both chemically abraded and untreated zircon show that both CA and untreated material can be
used interchangeably and dated against each other, so long as the corresponding reference value is used. Analyses of CA material are more precise, without any systematic discrepancies appearing at the 0.25-0.4% level. This suggests that for well-behaved reference zircons, using a piezo stage, automated analyses, and chemical
 abrasion, $^{206}$Pb/$^{238}$U precision substantially better than the 1-3% value quoted by Schaltegger et al. (2015) can be achieved without sacrificing accuracy.
We can test the dependence of the apparent high accuracy and precision of the CA-SHRIMP results on our choice of reference values by comparing them to a different set of internally consistent CA-ID-TIMS data. Huyskens et al. (2016) present CA-ID-TIMS results for TEMORA 2, OG1, and M16401 (the source rock for the 91500 zircon). By reducing our OGC and 91C results to the Huyskens et al. (2016) T2C value, we can look at accuracy & precision within the session and without intertracer uncertainties.
The Huyskens et al. (2016) reference values, calculated as for the other literature values from ratios using Student's t for uncertainty, are: 417.69 ± 0.47 Ma for TEMORA 2, 1064.97 ± 0.85 Ma for M16401, and 3463.49 ± 1.17 Ma for OG1.
Using this TEMORA 2 value of 417.69 Ma for T2C, the 91C and OGC values recalculated from session 170123 are: 1063.8 ± 3.0 for chemically abraded 91500 and 3456.8 ± 6.6 for chemically abraded OG1.




Reducing the data to the T2C values of Davydov et al. (2010), Ickert et al. (2015), or Von Quadt et al. (2016) instead of the
Schaltegger (2021) or Huyskens (2016) values does not significantly alter the results.

Chemical abrasion appears to ameliorate Pb loss at the scale of the 22μm x 16μm x 0.8μm sputter craters. SHRIMP ages of
chemically abraded OG1 and QGNG zircon are within uncertainty of the CA-ID TIMS ages, but not the untreated TIMS ages.
In contrast, the SHRIMP ages of untreated OG1 and QGNG (whether standardized to T2U or T2C) are within uncertainty of
the least discordant population of the TIMS results of untreated zircon, even after discordant grains are excluded to form a
coherent population (Figure 4). This supports the claim made both by Black et al. (2003) for QGNG and by Stern et al. (2009)
and Magee et al. (2023) for OG1, that the pooled untreated TIMS $^{206}$Pb/$^{238}$U age represents the $^{206}$Pb/$^{238}$U age of SHRIMP
spots on untreated material better than the $^{207}$Pb/$^{206}$Pb TIMS age (with or without chemical abrasion), or the chemically abraded
$^{206}$Pb/$^{238}$U TIMS age.

Bodorkos et al. (2009) suggested that sufficiently careful SHRIMP spot placement might avoid areas of Pb loss, while Magee
et al. (2016) showed (Supplementary figures DR12 and DR 13) that in Paleoarchean detrital zircons, 1 μm deep SHRIMP
spots show less Pb loss than 10-20 μm deep laser-ICPMS craters. However, the data presented here imply that there is a level
of subtle discordance that cannot be avoided in SHRIMP analyses by spot selection using transmitted, reflected, and
cathodoluminescence imaging. In other words, the dissolution of discordant zircon visible in the form of dissolved zones and
channels is not the only change to the zircon; the remaining visually intact material also undergoes a subtle change in U-Pb
ratio as a result of chemical abrasion. McKanna et al. (2023a) show that chemical abrasion removes material from the mm
scale all the way down to the submicron resolution of their SEM images, which is consistent with our SHRIMP U-Pb data.
The removal of pervasively distributed radiation damage confined to dislocation loops on the scale of 10 nm described by
Peterman et al. (2021) might account for this seemingly homogenous change on the submicron scale. At the much lower
annealing temperatures of chemical abrasion (900C vs 1450C), the closed system U-Pb behaviour suggests that U and Pb are
not diffusively separated. TEM and atom probe studies on chemically abraded zircon might yield insights on nanoscale
processes not spatially resolvable at the scale of a SHRIMP spot.

Use of chemical abrasion on the younger reference zircons 91500 and TEMORA 2 modestly reduces their scatter. For older,
slightly discordant reference zircons OG1 and QGNG, the reduction in analytical uncertainty from the CA treatment is more
substantial. Based on these results, we predict that the routine use of chemically abraded reference zircons could reduce the
reference zircon component of total uncertainty associated with SHRIMP U-Pb calibrations, even when analysing untreated
unknowns, as chemically abraded and untreated zircons appear to cross calibrate.

The 91500 age is unlikely to be significantly affected by the shallower ln(UO/U) vs ln (Pb/U) slope for 91500 zircon (figure
2) compared to the other reference zircons, as the ln(UO/U) values are only 0.4% lower for the 91500 grains than for the
TEMORA 2 CA reference zircons. For a difference in slope of 0.5, this would yield ages 0.2% older for both 91500 samples,
which is less than the analytical uncertainty for both the untreated and CA 91500 datasets.

It is worth noting that the source rocks for the TEMORA 2, QGNG, and OG1 reference zircons, the Middledale Gabbroic
Diorite (TEMORA-2), Quartz Gabbronorite Gneiss from the Donington Suite (QGNG), and the Owens Gully Diorite (OG1),
respectively, are all intermediate-to-mafic rocks. If any fractionation effects related to rock type exist, they are unlikely to be
perceived in this study. Zircon 91500 is derived from a porphyroblastic syenite gneiss (Wiedenbeck 1995). The shallower
calibration slope may relate to this, and its lower actinide, Hf, and REE contents which result from a different petrogenesis.

### 4.3 SHRIMP U- Pb of Mount Painter zircons

### 4.3.1 Igneous Rims

The analyses of the zircons from the Mount Painter Volcanics have several interesting features. Firstly, the chemically abraded
rims are much lower in common Pb than the untreated ones. The untreated rims have common $^{206}$Pb contents reaching as high
as 16% (Figure 11). The seven grains which have common $^{206}$Pb above 0.5% have been excluded from the weighted mean age,



but the common Pb correction works well enough for them to yield a coherent age of 430.3 ± 1.2 / 1.6 Ma, if included. This well-behaved common Pb correction suggests that the common Pb is close to the model age in composition, and is not Proterozoic industrial or environmental contaminant lead. If the Pb is contained in micro or nano inclusions, its presence in the rims might be explained by the rims on these volcanic zircons having crystallized rapidly and subsumed inclusions during

a volcanic eruption. The lack of common Pb in the chemically abraded zircons is consistent with the chemical abrasion process dissolving Pb-bearing mineral or melt inclusions less resistant to HF than zircon.

Such a hypothesis would predict that the rims on S-type granitic (not dacitic) zircons which crystalized more slowly would not contain as much common Pb. For comparison, eight ~430 Ma S-type granitic zircon overgrowths from Bodorkos et al. (2015) were re-examined to look for high common Pb. In three of these samples, no spots in the igneous overgrowth group had any

statistically significant common Pb. Three more had a single common Pb-containing outlier with a raw $^{207}Pb/^{206}Pb$ ratio of less than 0.075. The last two samples had, respectively, two and seven spots with detectable common Pb, but in no case was the total $^{207}Pb/^{206}Pb$ ratio higher than 0.075 (which is about 1.5% $^{206}Pb_c$ in the Devonian). So compared to these granitic S-type zircons of Bodorkos et al. (2015), our dacitic zircon rims have unusually high common Pb contents.

Unlike the reference zircons TEMORA 2, 91500, QGNG, and OG1, the MPC rims were not less scattered than the untreated

rims. The chemically abraded rims were more scattered. Whether or not the high common Pb rims were excluded, the MPU rims yielded a single population with a probability of fit (Pof) of either 0.11 or 0.3. In contrast, the chemically abraded zircons had a PoF below 0.001, even after excluding a spot which may have hit part of a core. This is not a case of the common Pb correction expanding the individual spot errors on the untreated grains to accommodate a similar level of dispersion.

Interestingly, the 18 chemically abraded cores which were the same age (within uncertainty) as the rims did not have excess

scatter. The 18 MPC cores all defined a single coherent population with a PoF of 0.13, and an age (431.3 ± 2.1 Ma) within uncertainty of the MPC rim age (431.8 ± 1.7 Ma). It is only the chemically abraded rims which show excess geochronological scatter.

The much larger population of syn-eruptive cores in the CA-treated grains as opposed to the untreated grains is interpreted as survivor bias, as the core-rim interface presents an area of weakness for the HF to attack during partial dissolution. Zircon

dissolution along this boundary was noted in some surviving grains (Figure 1).

The igneous ages from the MPC rims are about 1.5 to 2 million years older than those from the MPU rims. This 1/3 to ½ percent difference is consistent with the offset seen in TEMORA 2, QGNG, and OG1. While we have no self-annealing closure ages for QGNG and Mount Painter, Magee et al. (2017) give U-Th-He dates for OG1 and TEMORA that yield irradiation levels of up to $6.5x10^{17}$ α/g For TEMORA 2 and up to $2x10^{18}$ α/g for OG1 zircons. These are both over the $6x10^{17}$ α/g damage

limit proposed by McKanna et al. (2023b) where Pb loss may occur. The eruption age is unlikely to be the final cooling age for the Mt Painter Volcanics, as this unit was buried and deformed during the Lachlan Orogen. However, a closure time similar to TEMORA 2 (from the same orogen) would yield higher damage levels due to the higher uranium content of the Mt Painter zircon rims (tables 3 and 6b) relative to TEMORA 2.

The increase in scatter in the chemically abraded Mount Painter rims is more difficult to explain, and it occurs in both the

younger and older directions. Although Pb diffusion can happen in zircon at 950ºC, it is unlikely to proceed over tens of microns in the space of 15 hours. Even if the partial dissolution process enhances diffusion, the HF partial dissolution step comes after, not before, the high temperature annealing.

Given the number of high common Pb rims in the untreated rims (7 of 36), it is conceivable that the dissolution of included phases in the rims was incomplete, leaving orphaned U (in the case of the young grain), or orphaned Pb (in the case of the old

two).

Alternatively, the dissolution of inclusions and metamict zones in the rims may have compromised the sputtering surface. In theory, a nanospongiform surface texture from partial dissolution could cause ion emittance issues that might invalidate the $^{206}Pb/^{238}U$ vs $^{254}(UO)/^{238}U$ calibration equation. It is not clear why this would be apparent only in the Mount Painter zircons



and not the reference material grains, but it could be that the increased dissolution proportion of these grains has increased the
probability of any particular spot having anomalous $Pb^+$ - $U^+$ - $UO^+$ ion emission behaviour.

To constrain these various possibilities, the individual zircon grains whose rims did not group in the igneous age group were re-analysed (session 210046) to see if there was a reproducible difference in the $^{206}Pb/^{238}U$ ratio, or if the scatter was unreproducible. Both cores and rims were analysed to determine if the cores were higher or lower in total $^{206}Pb$ than the rims. The results, shown in Figure 12, show that the outlier grains from the original session 170124 are not consistently anomalous
in $^{206}Pb/^{238}U$ ratio or age. This implies an analytical effect relating to the sputtering and emission surface, not a compositional one due to Pb migration across zircon subdomains or orphaned U or Pb from incomplete dissolution.

It is worth noting that the SHRIMP instrument performed unusually well during reference zircon session 170123. The T2U grains formed a coherent population (probability of fit > 0.05) without the addition of any spot-to-spot error (Ludwig 2009). Currently in the Geoscience Australia SHRIMP laboratory, this only happens in approximately 20-30% of SHRIMP
geochronology sessions; The majority of sessions have a calculated spot-to-spot error between 0.5% and 1%, such as session 170124, in which the Mt Painter zircons were run (spot-to-spot error 0.66%). Continuing analytical development is necessary in order to understand the sources of this uncertainty. This may involve more rigorous control of ion source stability, instrument tuning, mount preparation, or stage movement and spot placement. However, chemical abrasion seems to ameliorate that component of spot-to-spot scatter caused by failure of untreated zircon to be completely closed to Pb mobilisation. What
instrumental issues remain the next largest source of uncertainty, how to ameliorate them, and where the ultimate limit of SIMS $^{206}Pb$-$^{238}U$ geochronology lies remains to be seen.

### 4.3.2 Mount Painter Volcanics cores

Compared MPU, MPC has more igneous age cores, and a smaller population of Ediacaran-Cambrian aged cores recognised as part of the Pacific Gondwana Suite (Fergusson et al. 2001). This is consistent with the Pacific Gondwana grains being more
susceptible to loss during the chemical abrasion procedure than other zircons in the sample. Similarly, the increase in the proportion of igneous age grains in the chemically abraded sample is consistent with single generation zircons being more resistant to destruction by the chemical abrasion process than those with overgrowths, where the contacts can be damaged by the partial dissolution step of chemical abrasion (figure 1) .

Geologically, the dominance of Pacific Gondwana Ediacaran-Cambrian zircons with a somewhat subordinate population of
1000-1200 Ma grains is typical of early Palaeozoic sediments in Queensland (Fergusson et al. 2002, 2007; Purdy et al. 2016), Victoria (Keay et al. 1999), South Australia (Ireland et al. 1998) and the NSW Lachlan fold belt to the South and East of Mount Painter (Fergusson et al. 2001). So their presence in the Mount Painter Volcanics is consistent with this unit being an S-type dacite sourced from the melting of early Paleozoic sediments (Chappell and White 1974). The relative dearth of ~450-480 Ma zircons in the Mount Painter Volcanics zircon cores may indicate derivation from Cambrian or early Ordovician sediments.
Alternatively, it may reflect sedimentary transport effects which deprived the source sediments of zircon younger than ~480 Ma.

### 4.4 SHRIMP $\delta^{18}O$ discussion

For the 91500, OG1, and QGNG zircons, the change in $\delta^{18}O$ mean values between the chemically abraded and unabraded populations was negligible. However, the chemically abraded TEMORA grains were approximately 0.7‰ lighter than the
untreated grains. This is probably a sample handling error. Schmitt et al. (2019) show that the TEMORA 2 zircon distributed by Geoscience Australia has variable $\delta^{18}O$ values, which differ depending on which batch of zircon is used. While the original plan was to split a single vial of TEMORA 2 zircon, a sample preparation miscommunication resulted in an entire vial being chemically abraded. Even though both the original and the replacement vials were labelled as being the older material (heavier in $\delta^{18}O$) described in Schmitt et al. (2019), we suspect an old vial was refilled with more recent material with lighter $\delta^{18}O$.



Otherwise, it is hard to explain why the TEMORA 2 zircon would change oxygen isotopic composition while the more metamict OG1 and QGNG samples remained unchanged. We repeat the warning from Schmitt et al. (2019) that TEMORA 2 oxygen isotope values from the Geoscience Australia mineral separation facility are heterogeneous.

The Mount Painter Volcanics zircon rims showed substantially more scatter in $\delta^{18}O$ composition than the reference zircons. This is consistent with previous observations that S-type granites have variable rim $\delta^{18}O$ (Ickert 2010). None-the-less, the

mean $\delta^{18}O$ value is consistent between the untreated and chemically abraded rims, and is well within uncertainty.

The Mount Painter Volcanics zircon cores have a wide range of $\delta^{18}O$ values, as might be expected of sedimentary zircons. For the dozen or so grains of each sample where core and rim analyses were performed on the same zircon, there is no evidence that the $\delta^{18}O$ composition of the core influences the $\delta^{18}O$ composition of the rim. Given the relatively low Ti-in-zircon rutile equilibration temperatures of 690°C (from both untreated and chemically abraded rims), this is not surprising. Similarly there

appears to have been no core-rim diffusive re-equilibration on the tens-of-micron spot diameter scale during the annealing phase of chemical abrasion.

It is worth noting that not all the Mount Painter Volcanics zircon cores are detrital. Those cores which are the same age (within uncertainty) as the rims are probably related to some sort of pre-eruptive igneous process. While the geochronology results show that the Th/U ratios of these cores are often higher than the rims, no other trace elemental analyses were done on the

cores. However, ten of these young cores were analysed for $\delta^{18}O$. These results were similar to the rims (between 7 and 10 ‰, with an average of 8.6 ±1 ‰), and were not within uncertainty of mantle $\delta^{18}O$ values.

**4.5 Trace element discussion**

While the OH/O ratio in zircons was measured in both the first (mass 16-17-18) and second (mass 17-18-19) SHRIMP SI sessions, the OH background was high, presumably due to the epoxy mounting material. This background decreased over time, so

the OH/$^{18}O$ data from the (17-18-19) session, which was run on the same mounts without any sample exchange, had lower backgrounds. The same trends were present in both sessions, but the lower signal / background ratio in the first (16-17-18) session increased the scatter.

Although we do not have a way of standardizing OH, the OH/O measurements in the untreated OG1 were both more variable and higher than in the chemically abraded OG1. In all other samples, there was no significant difference in OH between the

chemically abraded and untreated samples. This is consistent with OG1 being the most open system zircon we looked at, and thus having the most metamictisation-related hydration, which the partial dissolution step of chemical abrasion removes. Overall, the low OH/O in 91500 and S-type Mount Painter Volcanics cores relative to the I-type OG1, QGNG, and TEMORA 2 zircons is consistent with the observation of Mo et al. (2023) that S-type zircons are dryer than I-type zircons.

Fluorine contents in both the chemically abraded and untreated samples is unchanged. The lack of fluorine uptake in OGC is

consistent with hydrous material being preferentially dissolved, and not with the exchange of fluorine with OH bound in structurally sound zircon. Fluorine measured as $^{19}F^-$ by SHRIMP SI relative to $^{18}O^-$ and $^{19}F^+$ relative to $^{30}Si^+$ by SHRIMP 2 gave generally similar values, with the median values for all samples in the low teens of ppm, compared to a standard value of 15 ppm for 91500 (Coble et al. 2018).

For the other trace elements, little change was noticed. U and Th contents in the chemically abraded OG1 were slightly lower.

We attribute this to survivor bias, where higher U and Th grains would accrue more radiation damage, and be less likely to survive the chemical abrasion treatment. The Ti contents and indicated temperatures are unchanged. Burnham (2020) describes how to use the shape coefficients of O'Neill (2016)'s orthogonal polynomials to quantitatively describe REE concentrations. Median calculated shape coefficients are listed in Table 6, and spot by spot values are listed in supplementary tables 6 and 7. In the case of the laser trace element data, the requirement to also collect Pb isotopes has compromised the detection limits for

the LREE, and the lack of odd HREE in the run table means that the orthogonal polynomial calculations overfit the data due to the relatively few degrees of freedom caused by unanalysed odd HREE and BDL LREE. Thus laser lambda values are not



reported for the laser REE data. With the SHRIMP data, the lambda values for the curve defined by the median concentrations were fairly similar to the median lambda values as calculated for each spot for $\lambda_0$ and $\lambda_1$, vaguely similar for $\lambda_2$, and generally not well matched for $\lambda_3$.

The laser REE data show that for 4 of the 30 untreated QGNG and OG1 zircons, the chondrite-normalized lanthanum concentration was above 1. Additionally, in one of the OGC zircon, the chondrite-normalized lanthanum concentration was above 10. As lanthanum is highly incompatible in the zircon structure, these analyses probably indicate either inclusions or radiation damage associated alteration and element mobility. To the extent that they were above the detection limit, praseodymium and neodymium were also elevated in these samples. In contrast, none of the chemically abraded samples had

high lanthanum concentrations, indicating that these LREE-enriched zones present in the untreated samples were removed by the chemical abrasion process.

    None of the SHRIMP REE patterns had chondrite-normalized La concentrations above 1. We attribute this to the smaller analytical volume and shallower pit depth of SHRIMP relative to laser allowing the analyst to miss inclusions or metamict areas. The chemically abraded TEMORA 2 and OG1 zircons showed lower median LREE than the untreated samples,

consistent with the idea that a significant LREE budget in zircon comes from metamict zones or nanoinclusions of other minerals where the LREE are more compatible. However, QGNG shows the opposite trend, which we cannot explain.

### 4.6 Lu-Hf discussion

    The Hf isotopic results are similar for both chemically abraded and untreated reference zircons. Of the eight analysed reference zircons, only the initial Hf composition of the 91U was not within uncertainty of the reference values. There was no pattern in

the direction of Hf isotopic change between the chemically abraded and untreated samples, but the chemically abraded results generally had a larger uncertainty. QGNG (both) and OGU were the only reference zircons where the probability of fit for all spot measurements was greater than 5%.

### 5 Overall conclusions

    The most important conclusion from this study is that untreated and chemically abraded reference values and samples are not

interchangeable in SHRIMP U-Pb analyses, if they differ significantly. SHRIMPing chemically abraded zircons returns the chemically abraded TIMS age; SHRIMPing untreated zircons returns the TIMS age derived from zircons which did not undergo chemical abrasion. All chemically abraded reference zircon results in this study are more precise, but for the younger reference zircons (TEMORA 2 and 91500), this is a modest effect. Using CA-ID-TIMS reference ages for untreated older reference zircons such as OG1 or QGNG will almost certainly produce a systematic bias on the order of half a percent: the

difference between the reference ages for the untreated and chemically abraded grains observed using both SHRIMP and TIMS geochronology.

    This study shows that zircons such as OG1 and QGNG, which have suffered minor Pb loss, will not reproduce the CA-ID-TIMS age when analysed by SHRIMP, unless the SHRIMP targets are also chemically abraded. This suggests that there is pervasive low-level Pb loss in the untreated samples, which is corrected in the chemically abraded material. This would mean

that there is a subtle change in the surviving material, and it isn't just the visible channels and cavities (Figure 1) where zircon has been dissolved that are affected by the chemical abrasion process.

    As the SHRIMP ages of both untreated OG1 and untreated QGNG were about half a percent lower than the chemical abraded ages (either SHRIMP or TIMS), it would not be surprising if many older and/or actinide-rich zircons also showed minor Pb loss. This is consistent with the conclusions of Magee et al. (2023), where a comparison of untreated SHRIMP ages to CA-ID-

TIMS ages showed that Cambrian zircons tended to have younger SHRIMP ages, with a magnitude similar to the OG1 and QGNG offset demonstrated here.

This experiment demonstrates that when the SHRIMP instrument is running well, on chemically abraded reference zircons, accuracy and precision in the 2.5 to 4 ‰ range is possible. This represents a substantial improvement in the performance of in-situ geochronology. Whether this level of accuracy and precision will be achievable for unknown zircons with more complex geologic history or higher accumulated radiation damage than well-established reference zircons remains to be seen, as the Mount Painter Volcanics results suggest that there could be complicating factors. While we cannot guarantee that chemical abrasion will improve SIMS U-Pb geochronology, we think these results are promising enough to warrant further experimentation.

Chemical abrasion has little additional effect on zircon mineral chemistry beyond the U-Pb system, and does not seem to have compromised the ability to measure any of the elements or isotopes presented in this paper with microbeam mass spectrometry. OG1 showed a loss of OH, which is consistent with chemical abrasion dissolving altered, hydrated zircon. As this was not accompanied by an increase in F, we dismiss crystal structure F-OH substitution during HF treatment as a cause. Aside from OH, there was a tendency for highly incompatible elements such as the LREE, Ca, and common Pb to be reduced in the chemically abraded samples compared to the untreated samples, but this was not universal. Hafnium and oxygen isotopes were undisturbed.

## Author contributions

CM, SB, TI, and YA designed the project, CM and SB wrote the graduate project proposal for the project, SK did the gravimetric solution and OG1 ID-TIMS, CK and YA did the chemical abrasion for microanalysis, CK did SHRIMP analyses for U-Pb and $\delta^{18}$O, KW and NE did the LASS analyses, CM did the SHRIMP TE analyses, CK, CM, and NE reduced the data, CM, CK, YA, SK, and NE wrote the text, CK and CM did the figures, and CK, CM, SK, and NE did the tables.

## Competing interests

At least one of the co-authors is a member of the editorial board of Geochronology.

## Acknowledgements

We acknowledge the Ngambri and Ngunnawal nations, on whose lands the Geoscience Australia authors live and work. Geoscience Australia and all affiliated authors thank the Barngarla people for their assistance with land clearance and access, accompanying us to the site, and facilitating our QGNG-related research program. We thank David DiBugnara and the Geoscience Australia minerals separation facility for providing all the samples and making the mounts. We thank Andrew Cross, Keith Sircombe, and whomever the journal selects for thoughtful reviews. The Geoscience Australia Graduate Program enabled CK to pursue this project. GeoHistory Facility instruments were funded via an Australian Geophysical Observing System grant provided to AuScope Pty Ltd. by the AQ44 Australian Education Investment Fund program. The NPII multi-collector was purchased through ARC LIEF LE150100013. This paper is published with the permission of the CEO of Geoscience Australia.

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

**Figures:**

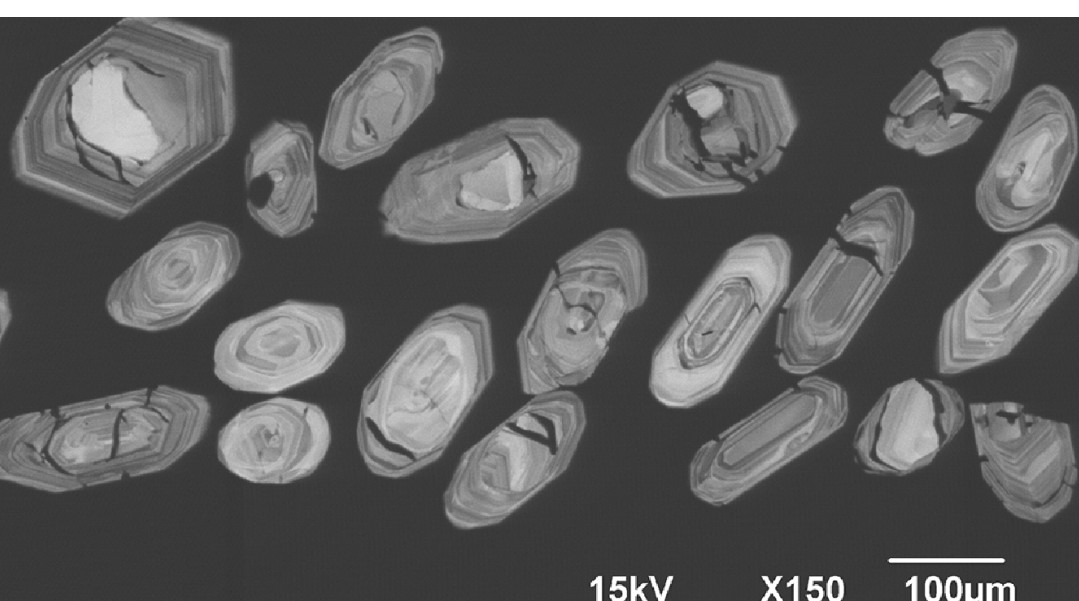

**Figure 1: CL image of chemically abraded Mount Painter Volcanics S-type dacitic zircons, showing etch channels from the partial HF dissolution. Many of these channels at least partially follow the core-rim boundary in those grains with both visible etch channels and inherited cores exposed at the level of the polishing plane.**





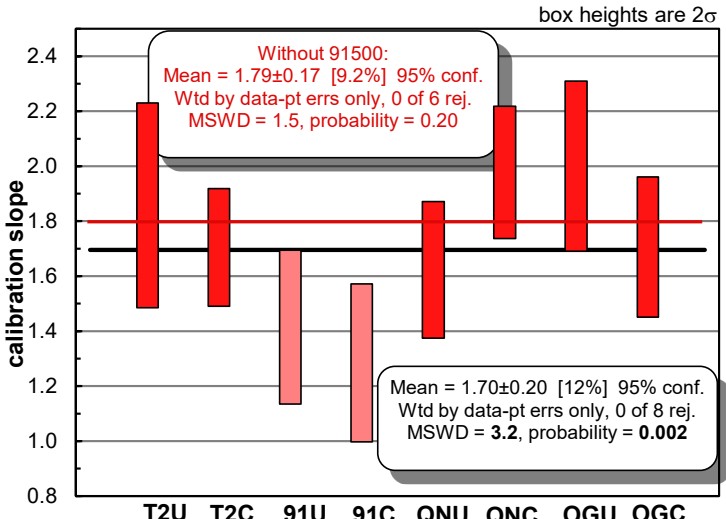

**Figure 2: The calibration slopes for all 8 samples on mount GA6363, when each is defined as the reference zircon. Number of analyses is 41 for T2C, 42 for all other samples.**





**Figure 3: The values of SHRIMP data for OG1, QGNG, 91500, and T2U, referenced to T2C. SHRIMP data are in red, reference TIMS ages are blue.**





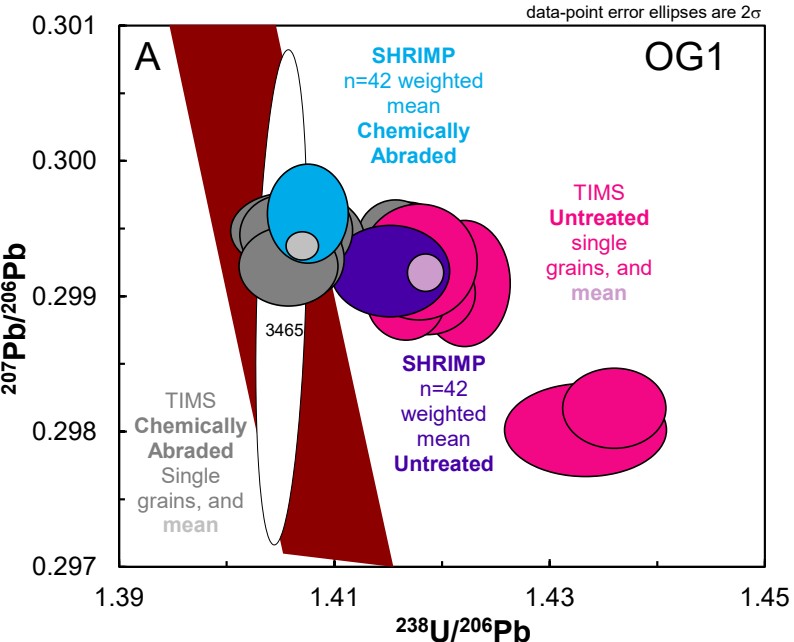


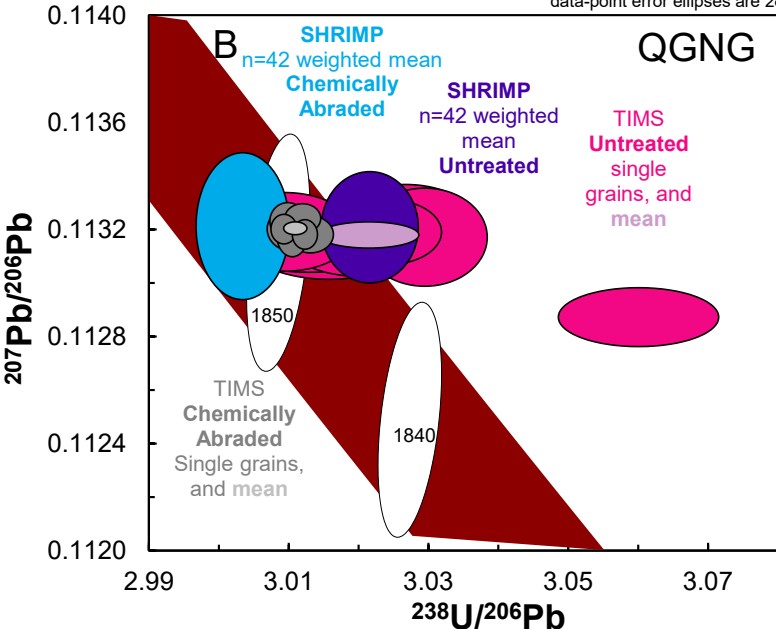

**Figure 4: Tera-Wasserberg concordia diagrams of OG1 (A) and QGNG (B) population weighted mean ages. Data are shown with internal errors only, to show the difference between the SHRIMP ages from the same analytical session. Data here standardised to T2U. TIMS analyses- both single grain and pooled ages- are shown for comparison, as are the single grain and pooled results for chemically abraded QGNG from Schoene et al. (2006). The Concordia band (with uncertainty) is brown with white oval age ticks.**




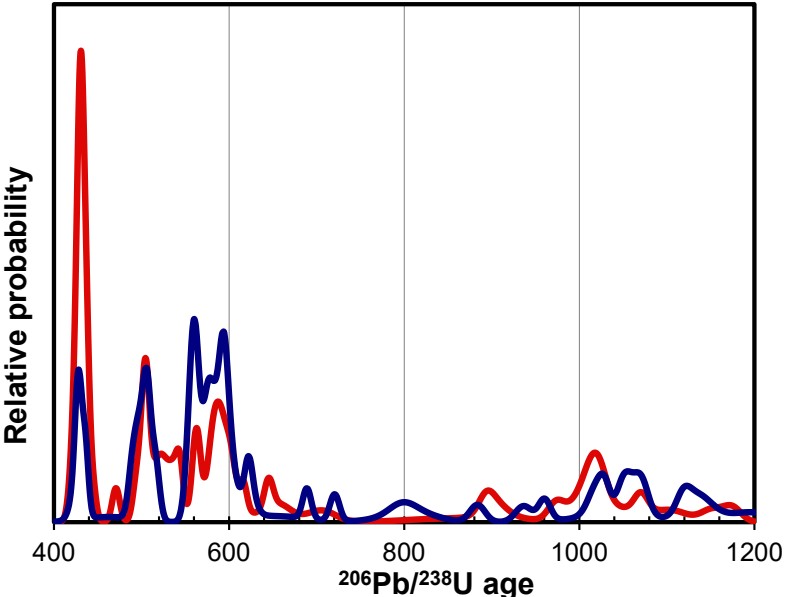

**Figure 5: Probability density diagram for inherited zircon core ages for zircons from the Mount Painter Volcanics. Red is chemically abraded. Blue is untreated. Grains older than 1200 Ma are not shown, as they are scattered individuals which do not form populations.**




**Figure 6: δ¹⁸O patterns for untreated and chemically abraded zircon populations. Untreated grains are green triangles; chemically abraded gains are blue diamonds. A. TEMORA 2. B: 91500. C: QGNG. D: OG1. E: Mount Painter Volcanic zircon rims. Uncertainties are 95% confidence intervals for weighted mean of all measurements in the population.**



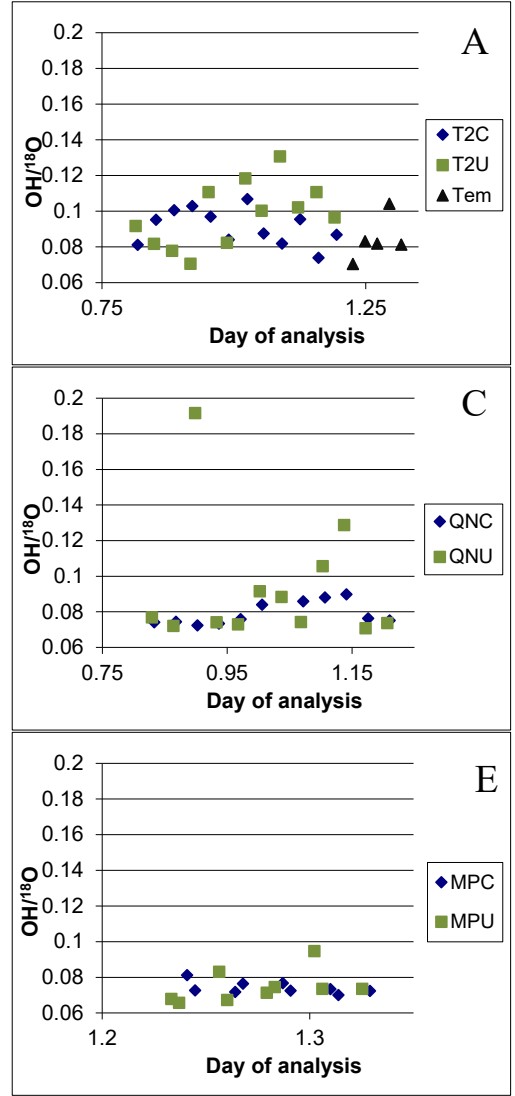

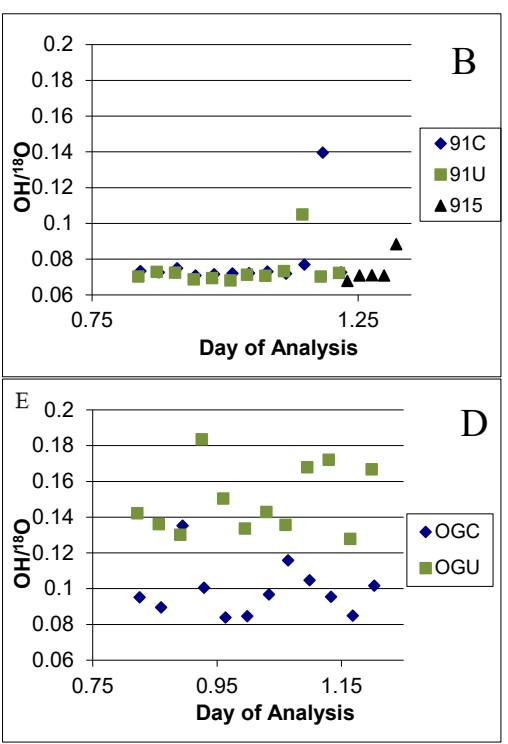


**Figure 7. OH/¹⁸O plots for various samples. All data have an epoxy degassing trend subtracted out. A: Untreated (green) and chemically abraded (blue) TEMORA 2. Black is untreated TEMORA 2 on the setup mount. B: Untreated (green) and chemically abraded (blue) 91500. Black is untreated TEMORA 2 on the setup mount. C: Untreated (green) and chemically abraded (blue) QGNG. D: Untreated (green) and chemically abraded (blue) OG1. E: Untreated (green) and chemically abraded (blue) Mount**
**Painter Volcanic zircon rims.**





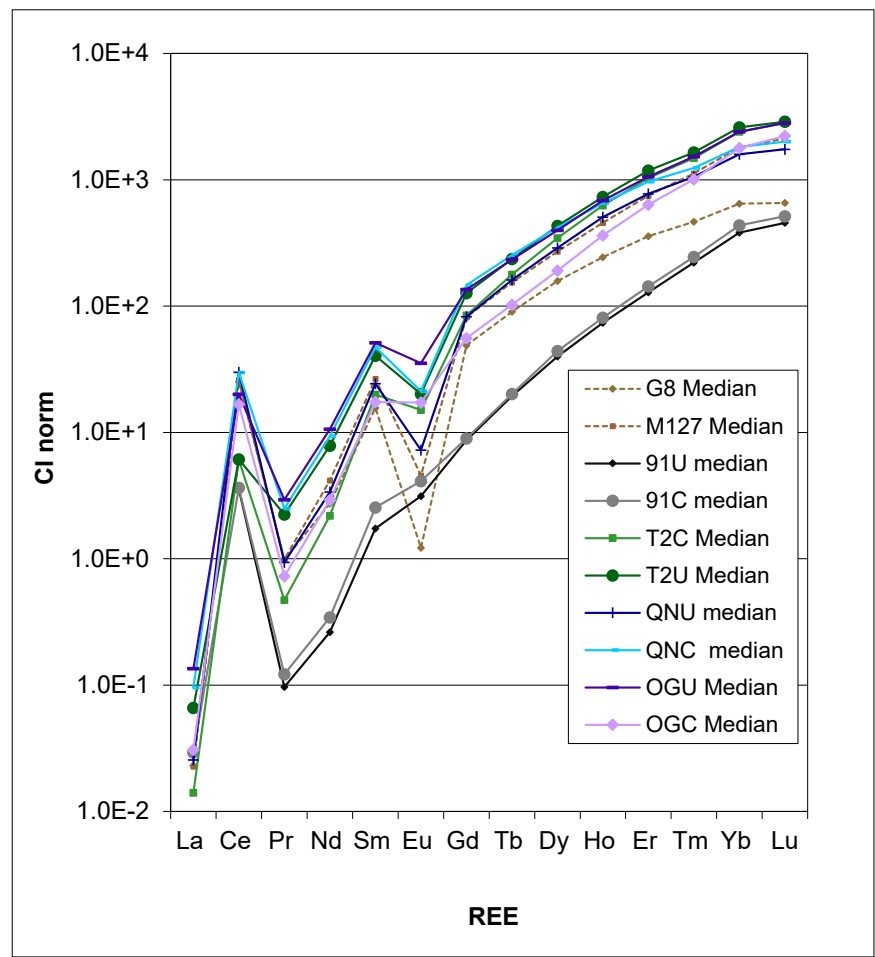

**Figure 8. REE plots for the median REE content of each reference zircon, normalized to chondritic abundances. Note that for OG1 and TEMORA, the LREE in the untreated samples are higher than in for the chemically abraded samples, while for QGNG the trend is reversed. 91500 shows little change. Secondary (untreated) reference zircons G8 and M127 also shown.**





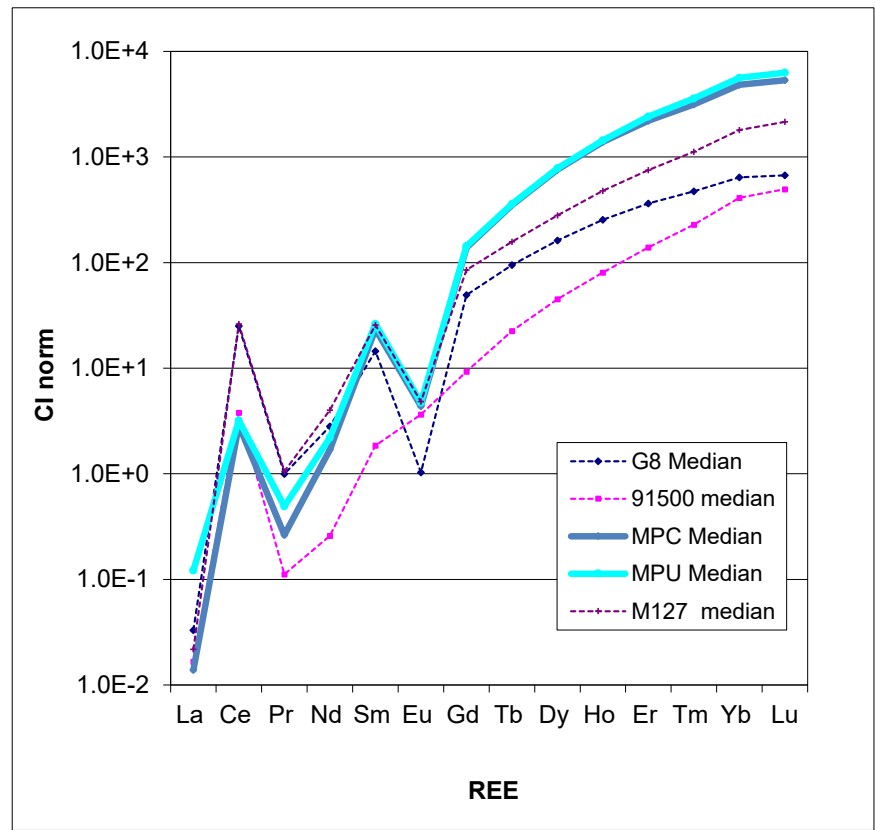

**Figure 9. REE plots for the median REE content of Mount Painter Volcanics zircon rims, normalized to chondritic abundances. Note that the untreated rims are higher in LREE than the treated ones. Secondary (untreated) reference zircons G8, 91500, and M127 also shown.**




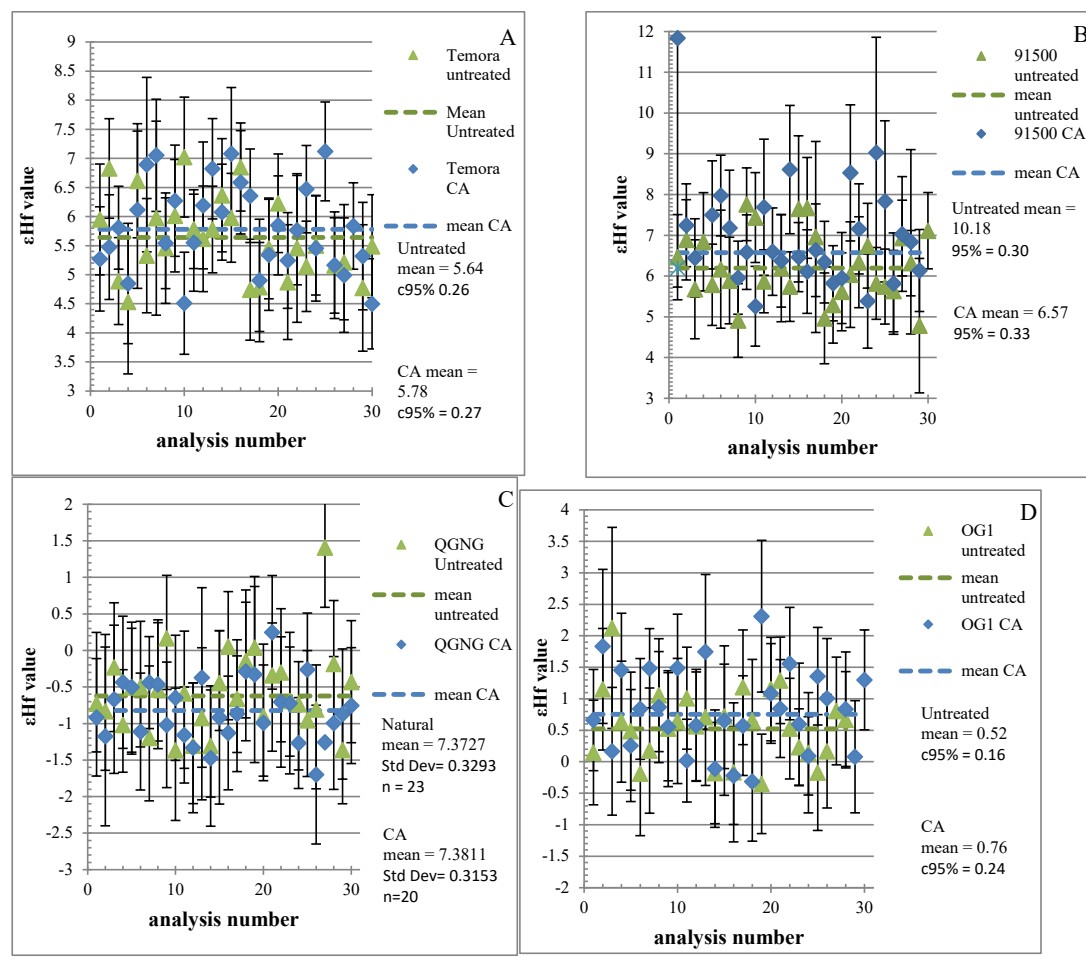

**Figure 10. Spot-by-spot Hf isotopic compositions of untreated and chemically abraded reference zircons. A: TEMORA 2; B: 91500; C: QGNG; D: OG1**





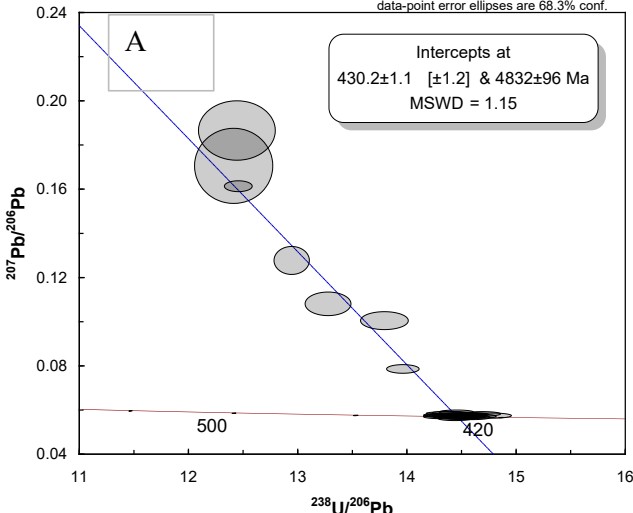


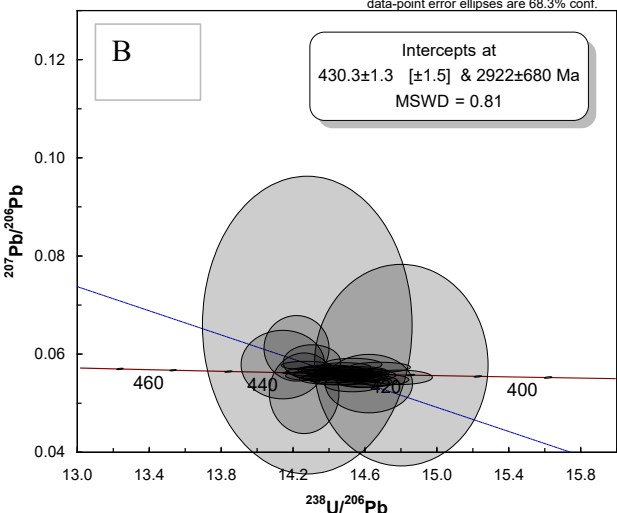

**Figure 11. Concordia diagrams for individual spot analyses for A. Untreated Mount Painter Volcanic zircon rims uncorrected for common Pb and B. The same analyses corrected for common Pb using $^{204}$Pb. Note the Y axis is almost twice as large in A than in B.**



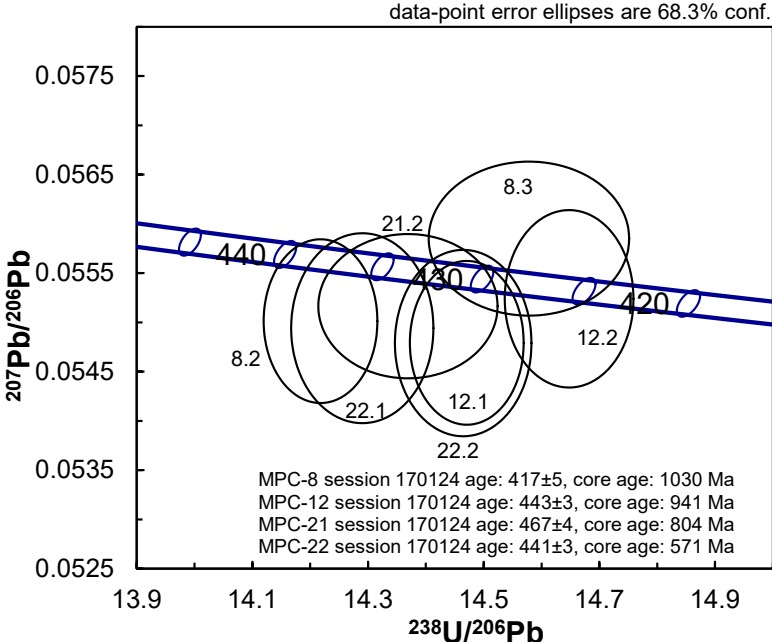

**Figure 12: Repeat analyses of Mount Painter Volcanics zircon rims with anomalous rim ages in the first analytical session (170124).**

**Tables:**

**Table 1: Reference values recalculated from literature using consistent uncertainty treatment and reduced tracer uncertainty from**
**gravimetric solution analyses:**

| Zircon | $^{206}Pb/^{238}U$ ratio | No tracer 95% * | Tracer 95% | Tracer+ random 95% | n | MS WD | Age Ma | ±No tracer 95% Ma | ±With tracer 95% Ma | Tracer | Reference |
|--------|--------|--------|--------|--------|----|------|--------|------|------|---------|------------|
| T2U  | 0.066789 | 0.098% | 0.051% | 0.278% | 9  | 0.56 | 416.78  | 0.39 | 0.44 | ROM     | Black et al. (2004)        |
| T2C  | 0.066896 | 0.012% | 0.030% | 0.044% | 59 | 3.89 | 417.36  | 0.05 | 0.13 | Both ET | Schaltegger et al. (2021)  |
| 91U  | 0.17937  | 0.025% | 0.030% | 0.039% | 7  | 0.58 | 1063.55 | 0.25 | 0.38 | ET535   | Schoene et al. (2006)      |
| 91C  | 0.17937  | 0.049% | 0.030% | 0.058% | 7  | 1.10 | 1063.51 | 0.49 | 0.57 | ET2535  | Horstwood et al. (2016)    |
| QNU  | 0.33097  | 0.229% | 0.051% | 0.347% | 8  | 2.94 | 1843.08 | 3.68 | 3.77 | ROM     | Black et al. (2003)        |
| QNC  | 0.33213  | 0.054% | 0.030% | 0.062% | 7  | 2.21 | 1848.70 | 0.86 | 0.99 | ET535   | Schoene et al. (2006)      |
| OGU  | 0.70503  | 0.125% | 0.051% | 0.301% | 6  | 1.46 | 3439.68 | 3.33 | 3.59 | ROM     | Stern et al. (2009)        |
| OGC  | 0.71074  | 0.106% | 0.051% | 0.340% | 7  | 0.22 | 3461.25 | 2.85 | 3.15 | ROM     | Stern et al. (2009)        |

\* 95% confidence interval is Std Er x Student's t x square root of MSWD, if probability of fit < 0.05.



**Table 2: SHRIMP session 170123 results normalized to A: T2U values of Table 1, and B: T2C values of Table 1. For natural OG1 and QGNG samples, where the data do not represent a single population, results are presented both for all data (italics, if not a coherent population) and the minimum number of rejections needed to bring the probability of fit above 0.05**

| A | Age Ma | Int 95% Ma | Ext 95% Ma | MSWD | Probability of Fit | n |
|---|---|---|---|---|---|---|
| T2C | 418.1 | 0.9 | 1.8 | 1.30 | 0.1 | 41 |
| 91U | 1067.8 | 3.3 | 5.0 | 1.07 | 0.35 | 42 |
| 91C | 1064.8 | 2.8 | 4.7 | 0.87 | 0.71 | 42 |
| *QNU* | *1841.8* | *4.4* | *7.5* | *2.23* | ***0.00*** | *42* |
| QNU | 1842.8 | 3.5 | 7.0 | 1.30 | 0.10 | 39 |
| QNC | 1852.6 | 3.3 | 7.0 | 1.30 | 0.11 | 42 |
| *OGU* | *3440.8* | *11.4* | *15.2* | *4.70* | ***0.00*** | *42* |
| OGU | 3445.8 | 5.9 | 11.7 | 1.41 | 0.058 | 34 |
| OGC | 3459.8 | 6.6 | 12.0 | 1.24 | 0.14 | 42 |

| B | Age Ma | Int 95% Ma | Ext 95% Ma | MSWD | Probability of Fit | n |
|---|---|---|---|---|---|---|
| T2U | 416.01 | 1.05 | 1.35 | 1.30 | 0.10 | 42 |
| 91U | 1065.9 | 3.1 | 3.8 | 1.07 | 0.34 | 42 |
| 91C | 1062.9 | 3.0 | 3.6 | 0.87 | 0.70 | 42 |
| *QNU* | *1838.7* | *4.4* | *5.6* | *2.23* | ***0.00*** | *42* |
| QNU | 1840.7 | 3.0 | 4.5 | 1.35 | 0.07 | 39 |
| QNC | 1849.4 | 2.9 | 4.5 | 1.27 | 0.11 | 42 |
| *OGU* | *3435.6* | *11.5* | *12.7* | *4.70* | ***0.00*** | *42* |
| OGU | 3440.6 | 6.0 | 8.2 | 1.41 | 0.06 | 34 |
| OGC | 3454.6 | 5.9 | 8.1 | 1.24 | 0.14 | 42 |





**Table 3: results of session 170123 using T2C as the primary reference zircon, calculated using all eight of the Jeon and Whitehouse (2014) calibration equations. Scattered or erroneous results are in grey:**

| Calibration type from Jeon & Whitehouse (2015) | 1σ error of mean % | MS WD | Prob. of fit | 1σ ext. spot-to-spot error | OGC Reference Age | Ref 95% | 204cor 206Pb/238U OGC Age | age error (95% conf.) | MS WD | PoF | QNC Reference Age | Ref 95% | 204cor 206Pb/238U QNC Age | age error (95% conf.) | MS WD | PoF |
|---|---|---|---|---|---|---|---|---|---|---|---|---|---|---|---|---|
| **206/238 vs 254/238** | 0.1 | 1.2 | 0.19 | 0 | 3461.2 | 3.2 | 3453.1 | 7.9 | 1.32 | 0.08 | 1848.7 | 0.9 | 1848.1 | 3.0 | 1.3 | 0.09 |
| 206/238 vs 270/238 | 0.11 | 1.4 | 0.06 | 0 | 3461.2 | 3.2 | 3454.0 | 9.8 | 1.47 | 0.03 | 1848.7 | 0.9 | 1846.6 | 3.1 | 0.9 | 0.72 |
| 206/238 vs 270/254 | 0.17 | 1.3 | 0.07 | 0 | 3461.2 | 3.2 | 3447.3 | 17.6 | 1.83 | 0 | 1848.7 | 0.9 | 1844.7 | 5.7 | 0.9 | 0.72 |
| **206/254 vs 254/238** | 0.10 | 1.2 | 0.2 | 0 | 3461.2 | 3.2 | 3453.4 | 8.0 | 1.38 | 0.06 | 1848.7 | 0.9 | 1848.2 | 3.0 | 1.3 | 0.09 |
| 206/254 vs 270/238 | 0.1 | 1.3 | 0.09 | 0 | 3461.2 | 3.2 | 3453.6 | 8.2 | 1.54 | 0.01 | 1848.7 | 0.9 | 1847.1 | 3.0 | 0.7 | 0.92 |
| 206/254 vs 270/254 | 0.12 | 1.4 | 0.07 | 0 | 3461.2 | 3.2 | 3456.0 | 9.9 | 1.48 | 0.03 | 1848.7 | 0.9 | 1846.4 | 3.5 | 0.6 | 0.99 |
| 206/270 vs 270/254 | 0.28 | 3 | 0 | 1.46 | 3461.2 | 3.2 | 3468.5 | 23.0 | 0.72 | 0.91 | 1848.7 | 0.9 | 1857.2 | 13.2 | 0.8 | 0.85 |
| 206/270 1D | 0.111 | 1.4 | 0.054 | 0 | 3461.2 | 3.2 | 3454.5 | 8.5 | 1.71 | 0 | 1848.7 | 0.9 | 1845.5 | 3.5 | 0.8 | 0.76 |

| Calibration type from Jeon & Whitehouse (2015) | 91C Reference age | Ref 95% | 204cor 206Pb/238U 91C Age | age error (95% conf.) | MS WD | PoF | 91U Reference age | Ref 95% | 204cor 206Pb/238U 91U Age | age error (95% conf.) | MS WD | PoF | T2U Reference age | Ref 95% | 204cor 206Pb/238U T2U Age | age error (95% conf.) | MS WD | PoF |
|---|---|---|---|---|---|---|---|---|---|---|---|---|---|---|---|---|---|---|
| **206/238 vs 254/238** | 1063.5 | 0.5 | 1061.8 | 2.9 | 0.79 | 0.8 | 1063.5 | 0.2 | 1064.9 | 3.1 | 1.1 | 0.31 | 416.78 | 0.44 | 415.84 | 1.08 | 1.3 | 0.08 |
| 206/238 vs 270/238 | 1063.5 | 0.5 | 1062.9 | 2.97 | 1.02 | 0.4 | 1063.5 | 0.2 | 1063.3 | 3.0 | 1.22 | 0.16 | 416.78 | 0.44 | 415.31 | 1.60 | 1.6 | 0.01 |
| 206/238 vs 270/254 | 1063.5 | 0.5 | 1061.2 | 7.0 | 1.58 | 0 | 1063.5 | 0.2 | 1057.7 | 5.6 | 1.12 | 0.27 | 416.78 | 0.44 | 414.71 | 2.23 | 1.3 | 0.13 |
| **206/254 vs 254/238** | 1063.5 | 0.5 | 1061.8 | 3.0 | 0.79 | 0.8 | 1063.5 | 0.2 | 1065.4 | 3.1 | 1.19 | 0.19 | 416.78 | 0.44 | 415.92 | 1.38 | 1.3 | 0.08 |
| 206/254 vs 270/238 | 1063.5 | 0.5 | 1062.8 | 2.6 | 0.6 | 1 | 1063.5 | 0.2 | 1064.9 | 2.9 | 1.07 | 0.35 | 416.78 | 0.44 | 415.53 | 1.31 | 1.4 | 0.07 |
| 206/254 vs 270/254 | 1063.5 | 0.5 | 1061.7 | 2.9 | 0.88 | 0.7 | 1063.5 | 0.2 | 1063.4 | 3.0 | 1.09 | 0.32 | 416.78 | 0.44 | 415.01 | 1.63 | 1.5 | 0.03 |
| 206/270 vs 270/254 | 1063.5 | 0.5 | 1072.2 | 7.8 | 1.21 | 0.2 | 1063.5 | 0.2 | 1067.3 | 7.7 | 1.29 | 0.1 | 416.78 | 0.44 | 415.48 | 3.45 | 1.2 | 0.23 |
| 206/270 1D | 1063.5 | 0.5 | 1062.5 | 2.8 | 0.96 | 0.5 | 1063.5 | 0.2 | 1062.7 | 3.3 | 1.21 | 0.17 | 416.78 | 0.44 | 415.23 | 1.4 | 1.3 | 0.09 |

**Table 4: $^{206}$Pb/$^{238}$U ages for samples on the Mount Painter Volcanics mount, standardized to untreated TEMORA 2 zircon (Black et al. 2004). Where the data do not represent a single population, results are presented both for all data (italics, if not a coherent population) and the minimum number of rejections needed to bring the probability of fit above 0.05**

| | n | Age (Ma) | int 95% (Ma) | Ext 95% conf (Ma) | MSWD | Probability of Fit | Rejections |
|---|---|---|---|---|---|---|---|
| OG1(untreated) | 18 | 3436.2 | 10.7 | 12.6 | 1.31 | 0.17 | 0 |
| 91500 (untreated) | 18 | 1060.1 | 5.9 | 6.4 | 1.02 | 0.43 | 2 |
| *MPC* | *36* | *431.8* | *1.7* | *2.0* | *1.97* | *0.0006* | *1* |
| MPC | 36 | 431.6 | 1.2 | 1.6 | 1.41 | 0.06 | 3 |
| MPU | 36 | 429.7 | 1.3 | 1.7 | 1.12 | 0.3 | 7 |
| MPC-young cores | 18 | 431.3 | 1.8 | 2.1 | 1.31 | 0.17 | 0 |
| MPU-young cores | 6 | 430.1 | 4.1 | 4.2 | 1.31 | 0.26 | 0 |
| *MPC-Core+Rim* | *54* | *431.6* | *1.3* | *1.7* | *1.72* | *0.0009* | *1* |
| MPC-Core+Rim | 54 | 431.3 | 1.0 | 1.5 | 1.23 | 0.13 | 4 |
| MPU-Core+Rim | 42 | 429.8 | 1.2 | 1.6 | 1.15 | 0.25 | 7 |

TEMORA 2 reference data: n=76; 1s er of mean=0.11%; MSWD=1.71; Probability of Fit=0.0001; spot-to-spot error=0.61%





**Table 5: Oxygen isotopic ratios and δ$^{18}$O weighted mean values for reference and Mount Painter Volcanics zircons.**

| Sample | Wt Mean $^{16}$O/$^{18}$O ratio | 95% conf | δ$^{18}$O | 95% conf | MSWD | probability of fit |
|--------|------------------|----------|-----------|----------|------|-------------------|
| T2U | 0.002030549 | 2.45E-07 | 8.20 | 0.12 | 7.91 | 2.28E-21 |
| T2C | 0.002029185 | 2.73E-07 | 7.53 | 0.13 | 9.03 | 7.54E-28 |
| 91U | 0.002034537 | 2.66E-07 | 10.17 | 0.13 | 11.24 | 6.81E-35 |
| 91C | 0.0020341 | 2.62E-07 | 9.95 | 0.13 | 7.28 | 3.67E-20 |
| QNU | 0.0020289 | 2.90E-07 | 7.38 | 0.14 | 12.94 | 1.58E-47 |
| QNC | 0.002028968 | 2.97E-07 | 7.42 | 0.15 | 9.47 | 3.24E-28 |
| OGU | 0.002026002 | 2.35E-07 | 5.95 | 0.12 | 6.94 | 2.70E-23 |
| OGC | 0.002025668 | 2.16E-07 | 5.79 | 0.11 | 5.60 | 3.66E-14 |
| MPU | 0.002032425 | 7.88E-07 | 9.13 | 0.39 | 43.22 | 1.02E-203 |
| MPC | 0.002033039 | 9.54E-07 | 9.43 | 0.47 | 58.00 | 8.93E-222 |





**Table 6. SHRIMP median sample trace elemental concentrations measured as positive ions on SHRIMP 2 and median derived values. A: Session 220029, on the reference zircon mount GA6363. B: Session 220028, on Mount Painter Volcanics mount GA6364. "Setup 91500" is co-analysed 91500 zircon on setup mount GA5040. T (Ti): Titanium-in-zircon thermometer of Ferry and Watson (2007), assuming titanium activity=1. P sat: Phosphorus saturation of Burham and Berry (2017). Orthogonal polynomials (λ) of O'Neill (2016).**

| A | G8 | M127 | T2U | T2C | 91U | 91C | QNU | QNC | OGU | OGC |
|---|---|---|---|---|---|---|---|---|---|---|
| F ppm | 46 | 15 | 15 | 14 | 16 | 17 | 14 | 15 | 16 | 16 |
| Al ppm | 1.5 | 3.9 | 1.9 | 4.1 | 11.8 | 9.7 | 0.1 | 0.1 | 1.9 | 2.2 |
| P ppm | 81 | 190 | 128 | 193 | 11 | 10 | 259 | 220 | 166 | 127 |
| Ca ppm | 2.3 | 2.6 | 2.5 | 2.5 | 2.5 | 3.5 | 2.5 | 1.6 | 2.5 | 3.2 |
| Ti ppm | 9.5 | 5.8 | 9.6 | 7.7 | 4.4 | 4.6 | 15.6 | 14.4 | 8.1 | 6.3 |
| Fe ppm | 4.2 | 2.1 | 1.1 | 4.0 | 3.7 | 3.1 | 5.1 | 4.5 | 38.1 | 55.5 |
| Y ppm | 456 | 780 | 1186 | 1037 | 121 | 132 | 809 | 1033 | 1142 | 635 |
| | | | | - | | | | | | |
| La ppm | 0.005 | 0.005 | 0.016 | 0.003 | 0.005 | 0.007 | 0.006 | 0.023 | 0.032 | 0.007 |
| Ce ppm | 15.4 | 16.0 | 3.7 | 3.7 | 2.2 | 2.2 | 18.3 | 18.2 | 12.3 | 10.2 |
| Pr ppm | 0.091 | 0.092 | 0.208 | 0.044 | 0.009 | 0.011 | 0.087 | 0.232 | 0.273 | 0.068 |
| Nd ppm | 1.2 | 1.9 | 3.6 | 1.0 | 0.1 | 0.2 | 1.5 | 4.3 | 4.8 | 1.3 |
| Sm ppm | 2.3 | 3.9 | 6.0 | 3.0 | 0.3 | 0.4 | 3.6 | 7.0 | 7.6 | 2.6 |
| Eu ppm | 0.07 | 0.26 | 1.13 | 0.85 | 0.18 | 0.23 | 0.41 | 1.19 | 1.98 | 0.97 |
| Gd ppm | 9.7 | 16.2 | 25.3 | 16.8 | 1.7 | 1.8 | 16.4 | 29.2 | 27.0 | 11.1 |
| Tb ppm | 3.3 | 5.6 | 8.5 | 6.4 | 0.7 | 0.7 | 5.8 | 9.2 | 8.4 | 3.7 |
| Dy ppm | 39 | 67 | 106 | 85 | 10 | 11 | 71 | 103 | 98 | 47 |
| Ho ppm | 13 | 25 | 40 | 34 | 4 | 4 | 28 | 36 | 37 | 20 |
| Er ppm | 57 | 120 | 189 | 165 | 21 | 23 | 124 | 154 | 169 | 102 |
| Tm ppm | 12 | 28 | 41 | 37 | 5 | 6 | 26 | 31 | 38 | 25 |
| Yb ppm | 104 | 291 | 418 | 387 | 62 | 70 | 257 | 292 | 386 | 288 |
| Lu ppm | 16 | 52 | 71 | 69 | 11 | 13 | 43 | 49 | 69 | 55 |
| Hf ppm | 11351 | 11472 | 7960 | 8668 | 5388 | 5233 | 10196 | 9977 | 8261 | 8759 |
| Th ppm | 254 | 397 | 78 | 118 | 23 | 23 | 128 | 207 | 134 | 86 |
| U ppm | 1349 | 851 | 156 | 283 | 70 | 71 | 166 | 230 | 138 | 150 |
| | | | | | | | | | | |
| T (Ti), °C | 741 | 697 | 743 | 722 | 673 | 679 | 790 | 782 | 727 | 705 |
| P sat | 0.34 | 0.40 | 0.17 | 0.30 | 0.14 | 0.11 | 0.61 | 0.33 | 0.23 | 0.38 |
| | | | | | | | | | | |
| $\lambda_0$ | 3.2 | 3.8 | 4.3 | 3.7 | 1.8 | 2.0 | 3.7 | 4.3 | 4.4 | 3.6 |
| $\lambda_1$ | -53 | -58 | -55 | -65 | -58 | -56 | -58 | -51 | -51 | -58 |
| $\lambda_2$ | -216 | -202 | -176 | -202 | -32 | -2 | -206 | -197 | -150 | -130 |
| $\lambda_3$ | -1161 | -1669 | -1384 | -692 | 590 | 720 | -996 | -1078 | -1379 | -1375 |




| B | G8 | 91500 | 91500 Setup | M127 | MPU | MPC |
|---|---|---|---|---|---|---|
| F ppm | 47 | 16 | 15 | 15 | 16 | 16 |
| Al ppm | 1.8 | 12.5 | 12.7 | 4.6 | 38.3 | 31.5 |
| P ppm | 83 | 12 | 12 | 194 | 1142 | 1075 |
| Ca ppm | 2.4 | 2.3 | 1.9 | 1.7 | 6.0 | 2.6 |
| Ti ppm | 9.3 | 4.6 | 4.7 | 6.0 | 5.3 | 5.3 |
| Fe ppm | 5.5 | 6.6 | 3.6 | 3.0 | 8.4 | 3.9 |
| Y ppm | 451 | 125 | 130 | 804 | 2286 | 2188 |
| La ppm | 0.008 | 0.004 | -0.003 | 0.005 | 0.029 | 0.003 |
| Ce ppm | 15.3 | 2.3 | 2.3 | 16.1 | 2.0 | 1.8 |
| Pr ppm | 0.092 | 0.010 | 0.005 | 0.098 | 0.046 | 0.024 |
| Nd ppm | 1.3 | 0.1 | 0.1 | 1.8 | 1.0 | 0.8 |
| Sm ppm | 2.2 | 0.3 | 0.3 | 3.8 | 3.9 | 3.5 |
| Eu ppm | 0.06 | 0.20 | 0.21 | 0.27 | 0.27 | 0.25 |
| Gd ppm | 9.8 | 1.8 | 1.9 | 17.0 | 28.5 | 27.6 |
| Tb ppm | 3.4 | 0.8 | 0.8 | 5.7 | 12.9 | 12.7 |
| Dy ppm | 40 | 11 | 10 | 69 | 193 | 187 |
| Ho ppm | 14 | 4 | 4 | 26 | 79 | 76 |
| Er ppm | 58 | 22 | 23 | 120 | 386 | 353 |
| Tm ppm | 12 | 6 | 6 | 28 | 89 | 78 |
| Yb ppm | 103 | 66 | 65 | 289 | 906 | 776 |
| Lu ppm | 16 | 12 | 12 | 53 | 155 | 131 |
| Hf ppm | 11252 | 5301 | 5512 | 11602 | 11278 | 11029 |
| Th ppm | 249 | 26 | 27 | 406 | 70 | 64 |
| U ppm | 1311 | 70 | 74 | 843 | 442 | 398 |
| | | | | | | |
| T (Ti), °C | 740 | 678 | 679 | 700 | 690 | 690 |
| P sat | 0.35 | 0.14 | 0.13 | 0.42 | 0.85 | 0.86 |
| | | | | | | |
| $\lambda_0$ | 3.3 | 1.9 | 1.6 | 3.8 | 4.0 | 4.4 |
| $\lambda_1$ | -52 | -58 | -66 | -58 | -72 | -64 |
| $\lambda_2$ | -195 | -29 | -149 | -206 | -211 | -80 |
| $\lambda_3$ | -792 | 747 | -389 | -1662 | 779 | 1920 |




**Table 7. Summary of laser ICPMS U-Pb results.**

| Sample | $^{206}Pb/^{238}U$ ratio | c95% (in %) | MSWD | Probability of fit | age (Ma) | 95% conf (Ma) |
|---|---|---|---|---|---|---|
| T2U | 0.066273 | 1.226 | 1.71 | 0.01 | 413.7 | 5.1 |
| T2C | 0.065788 | 0.366 | 1.34 | 0.10 | 410.7 | 1.5 |
| 91U | 0.176516 | 0.854 | 1.62 | 0.02 | 1047.9 | 8.9 |
| 91C | 0.180794 | 1.312 | 3.32 | 0.00 | 1071.3 | 14.1 |
| QNU | 0.334375 | 0.547 | 2.83 | 0.00 | 1859.6 | 10.2 |
| QNC | 0.331048 | 0.436 | 1.76 | 0.01 | 1843.5 | 8.0 |
| OGU | 0.713947 | 0.471 | 1.96 | 0.00 | 3473.3 | 16.4 |
| OGC | 0.707858 | 0.559 | 2.80 | 0.00 | 3450.4 | 19.3 |

**Table 8: Summary of laser ICPMS Hf isotopic results:**

| Sample | Initial $^{176}Hf/^{177}Hf$ | 95% conf | MSWD | PoF | n | εHf(t) | 95% | Ref Value | Reference |
|---|---|---|---|---|---|---|---|---|---|
| T2U | 0.2826821 | 7E-06 | 2.3 | 0 | 28 | 5.64 | 0.26 | 5.8 | Woodhead et al. 2005 |
| T2C | 0.282686 | 8E-06 | 2.7 | 0 | 30 | 5.78 | 0.27 | 5.8 | Woodhead et al. 2005 |
| 91U | 0.282287 | 8E-06 | 2.3 | 0 | 29 | 6.19 | 0.30 | 6.9 | Woodhead et al. 2005 |
| 91C | 0.2822977 | 9E-06 | 2.1 | 0 | 29 | 6.57 | 0.33 | 6.9 | Woodhead et al. 2005 |
| QNU | 0.2815866 | 4E-06 | 1.4 | 0.09 | 30 | -0.62 | 0.15 | -0.7 | Woodhead et al. 2005 |
| QNC | 0.281581 | 4E-06 | 0.9 | 0.57 | 30 | -0.82 | 0.15 | -0.7 | Woodhead et al. 2005 |
| OGU | 0.2805553 | 5E-06 | 1.3 | 0.12 | 29 | 0.52 | 0.16 | 0.6 | Kemp et al. 2017 |
| OGC | 0.2805618 | 7E-06 | 2.2 | 0 | 29 | 0.75 | 0.24 | 0.6 | Kemp et al. 2017 |