# Peer review of "Effect of chemical abrasion of zircon on SIMS U-Pb, $\delta^{18}$ O, trace element, and LA-ICPMS trace element and Lu-Hf isotopic analyses"

_EGUsphere, 2023_

## Author Comment (AC1)

Response to review #1:

We appreciate Reviewer #1's interest in our project, and regret that they missed the main points of our paper, which we repeat below for clarity:

The main point of our paper is that chemically abraded reference ages should not be used for non-chemically abraded (untreated) reference zircons for SIMS U-Pb analysis. We reproduce chemically abraded CA-ID-TIMS reference ages for chemically abraded SIMS targets, and untreated ID-TIMS reference ages for untreated SIMS targets. This is the most important and urgent part of this paper, as many SIMS analysts currently use CA ages for natural material. While the differences for most reference zircons are too small to distinguish in an individual session, this will produce systematic errors over time for any reference zircon where these ages appreciably differ.

Our secondary point, and the reason the study was conducted, is to see if the chemical abrasion process interferes with measurements of other chemical and isotopic systems, either by mobilizing elements, or by altering the matrix in a way that interferes with the analysis. Our conclusions are that this does not happen.

Our tertiary point is that if SIMS U-Pb analysts wish to achieve sub-percent accuracy and precision, chemical abrasion will be necessary, but not sufficient, for all but the least radiation-damaged zircons. Natural zircon is not a reliably closed system at sub-percent levels. Once the open system material is removed by chemical abrasion, highly accurate and precise ages are possible, but are not yet routine.

We admit that due to the large amount of data in this paper has resulted in us plotting only the barest minimum of figures to make our point; perhaps we have been too miserly in this regard. We can happily produce the additional figures that reviewers request, with the caveat that this would lengthen what is already a long paper.

Should the editor prefer the paper be split into multiple papers (e.g. one of TIMS and SIMS U-Pb, and one on the rest of the periodic table and Hf and O isotopic systems), we are open to such an approach, however we would like to point out that reviewer's #1's desire to see this work both published soon and completely rewritten is self-contradictory.

Reviewer #1 engages in excessive speculation regarding the zircons from Mt Painter Volcanics, a S-type dacite in the Lachlan Fold Belt, Australia. We admit an oversight in representing the literature on this topic, the inclusion of which may answer some of reviewer #1's questions.

While we were preparing this manuscript for submission we failed to notice the publication of Vogt et al. (2023), which describes the chemical abrasion and subsequent SIMS analysis of S-type granitic zircons from Central Europe.

Like the study of Kryza et al. (2012), Vogt et al. (2023) chemically abrade only their unknown sample, and do not look at the effect of CA on well-characterised zircons. However, Vogt et al. (2023) provide SIMS U-Pb, $\delta^{18}O$, and trace element results for both untreated and chemically abraded samples which answer many of Reviewer #1's speculations on this topic.

We feel that that the very different aims of our study and Vogt et al.'s (2023) should make explaining their complementarity fairly straightforward. However, if the editors feel that Vogt et al. (2023) renders our S-type work redundant, and that this work is confusing to readers like Reviewer #1, we suppose we could remove it for clarity and publish just the reference zircon data.

With regards to the scatter of this data, almost all SIMS U-Pb data are scattered beyond the statistical expectations of the grouped spots. This is why it was noteworthy that the chemically abraded reference zircon analytical session (170123) did not require any spot-to-spot error component. Thus, the scatter in the MPC data is no different than the vast majority of SIMS U-Pb data collected over the last 40 years. This is why we don't dwell on it, aside from noting that it shows that chemical abrasion is not a magic bullet that will eliminate spot-to-spot error every time in every sample of every session. The outliers were worth checking to make sure there was no reproducible change that might indicate diffusion (or some other elemental mobility) induced by the CA process, but outlier spots are not that unusual. To put them in perspective, with the exception of spot MPC.21.1 (which clipped the tip of a core), the rest of the spots all group with a spot-to-spot error of 1.02%. While slightly higher than our lab's long term average, this is quite good by historical standards (see figure 1 of Magee et al. 2024 for context).

Similarly, CA-ID-TIMS data also often fail to group. Of the 36 TIMS results examined in Magee et al. (2023), only 13 had a single group. While many of these (particularly the Permian volcanics) are obviously geological (as shown in Metcalfe et al. 2015 and Laurie et al. 2016), rejection of outliers is commonplace in the CA-ID-TIMS literature. Outlier treatment in SIMS is a large and contentious topic which we feel is beyond the scope of this paper; this is why we provide both outlier rejected and included data for those samples which do not group (OGU, QNU, and MPC) in tables 2 and 3.

Finally, if there is a SIMS surface effect created by chemical abrasion, it is of a smaller magnitude than the variation of U-Pb measurements caused by open system behaviour in untreated zircon. The difference between MPC and MPU is slightly less than half a percent. However, Magee the al. (2024) show that SIMS dates of untreated zircon can be as much as 2% different to their CA-ID-TIMS dates, due to open system behaviour in the zircon. So even if we are introducing a new source of bias, it is several times smaller than the open system problem which it solves.

We do not use $^{207}$Pb common Pb correction for the age of the Mount Painter Volcanics because publishing their age is not the point of the paper. The point of the paper is to examine the effect of chemical abrasion on zircon. Obviously multi-age domain zircons are never used as reference zircons, so Mount Painter Volcanics zircons were picked because they were, in our opinion, the most likely multi-age zircon we had in our collection that we thought would behave well (As we state in section 1.2). Comparing zircons requires similar data treatment for all samples. And $^{207}$Pb corrections cannot be performed on OG1 because around ~3450Ma, the common Pb correction line and the concordia curve are parallel, so tiny deviations from concordance produce huge changes in age (and reverse concordant data has no intercept at all!). Thus for consistency we do not use $^{207}$Pb corrections on any samples.

With regards to the many comments on S-type zircons, the applicability of the S-type label to rocks beyond the Silurian of the Lachlan Fold Belt is well beyond the scope of this paper. We feel that data on LFB ~430 Ma S-type granites (e.g. those of Ickert 2012 or Bodorkos et al 2015 ) are the best approximation for what to expect from the Mount Painter Volcanics, a ~430 Ma S-type pyroclastic dacite. Again, the purpose of this sample in this study is to determine of chemical abrasion does unexpected things to multi-stage zircons, and as we state in section 1.2, zircons from the Mount Painter Volcanics were chosen as the simplest example of a zircon with multiple growth domains.

Addressing Reviewer #1's specific points:

We are well aware of the McKanna et al. (2023) study, as I (CM, the corresponding author) provided a community comment during the review process. It is complementary to our work, as their imaging

of ever-smaller channels of dissolution down to the resolution of their instrument is consistent with our observation that the removal of low level Pb loss is pervasive at the scale of our SHRIMP spot.

We do not maintain the argument that chemical abrasion of moderately damaged (or highly dissolved) zircon produces worse SIMS results because neither the literature nor our data support that argument. Previous studies (Kryza et al. (2012), Watts et al. (2016), and Vogt et al. 2023) between them showed improvement in all 5 of their unknowns. Our data for OG1 and QGNG similarly show vast improvements. Whether the MPC data is better or worse depends on one's outlier treatment philosophy and data quality metrics; MPC is certainly lower in common Pb than MPU, and the outlier-rejected age is more precise. But with 4 studies showing that 7 zircons give better CA data, 4 reference zircons (91500, R33, Temorax2) are unchanged within uncertainty, and only one might be worse depending on how you treat the data, to catastrophize about the use of CA for SIMS is nonsensical.

Reviewer 1's speculation about secondary ion emission behaviour due to the change in texture from large scale dissolution is interesting, but tangential. Should the editors feel it is important to address, we have data from follow-up experiments which show that OG1 chemically abraded with a more aggressive recipe produces results that are at least as good, if not better that the results published here, and with a much higher proportion of dissolution channels. We did not include that work in this paper because this study is long enough already, and it is a minor secondary concern. Similarly, we have hundreds of sessions worth of OG1 secondary standard data we could present to show that the SIMS age reproduces the untreated (and not the CA) ID-TIMS $^{206}Pb/^{238}U$ age, but we didn't feel the need to belabour this point (Which was previously made in Black et al. (2003), Stern et al. (2009), and Magee et al. (2023)).

Reviewer 1's suggestion that the difference in igneous age cores between MPU and MPC is due to selection bias is backwards: inherited cores are easier to see in MPC, so it is less likely that we would hit igneous ones by accident while targeting detrital cores from the protolith, as was our stated goal (line: 256-257). All analyzed spots are annotated in the supplementary figure 2 if the reviewer is curious about spot placement. There is no need to speculate.

Reviewer 1's point about the radiation damage history of our reference zircons (sample selection) is valid. Sadly, not all reference zircons are completely characterized with regards to the time at which they cooled through their self-annealing temperature limit (lines 97-100). Our main aim was to find materials with a variety of ages which had both chemically abraded and untreated reference values. While damage estimates for OG1 and Temora-2 were published in Magee et al. (2017), we are not aware of any determinations for QGNG. Regional constraints suggest it cooled through ~300C (K-spar argon closure from Foster and Ehlers 1998) around 1600 Ma, and through apatite fission track closure (~100C) around 250Ma (Kohn et al. 2002). These provide uncertainty of more than a billion years for when QGNG dropped below the self-annealing temperature. None-the-less, most of central Australia has similarly nebulous cooling constraints as QGNG, so we think it is a useful reference material to investigate, as much of our work is done in the central Australian Paleoproterozoic.

Reviewer 1's comment about internal and external error is misplaced. Internal error is to be used when comparing results run from within the same analytical session, which is most of what we do in this paper. External error is also reported and used in relation to comparing our SIMS results to reference values.  This is why we took the time in the introduction to carefully explain this difference.

 Reviewer 1's comments on calibration slopes is confusingly written. Any of the eight reference zircon samples can be used as a primary standard to produce a calibration line. We mostly present data

using T2U or T2C as the primary reference materials for convenience. For each reference zircon, the calibration slope of the untreated and CA samples are within uncertainty of each other, as is shown in figure 1. 91500 (both untreated and CA) has a different slope than the other three. As we specifically state in the paper (line 400) the only permutation of references and unknowns which isn't within analytical uncertainty is the OGC-91U pair, and even these are only mismatched by less than half a percent.

Replies to the specific line-by-line comments:

L149. These CA-ID_TIMS OG11 results were delivered to GA too late for inclusion in Stern et al. 2009, and instead presented in poster form by Bodorkos et al. 2010. We are presenting them in full here in supplement table 1.

L502. Uncertainties including tracer are always larger than without the tracer, because tracer uncertainties are non-zero. However, there is an error in table 1, in that the values tracer+random 95% on the ratio did not recalculate when our new tracer uncertainty was applied. We thank the reviewers for making us check this.

L582-3. The Mount Painter U-Pb data are shown in table 3 and supplementary table 5.

L613 Relative Rim-Core $^{206}$Pb concentration must be known to predict the direction of any potential diffusion.

L617: We are saying that chemical abrasion alone will not eliminate spot-to-spot error. There are still session-specific instrumental effects, not all of which we have solved, which can and do occur.

L627 It isn't a guess to say that 18 is larger than 6, or that 12 is smaller than 22.

L657 Rims on low $\delta^{18}$O cores having their $\delta^{18}$O values dragged down relative to the rim average (or vice versa for high $\delta^{18}$O cores) would be evidence of core-rim oxygen exchange. We do not see this, but it would be hard to see due to the heterogeneity in rim $\delta^{18}$O in this sample.

L658 Ti-is zircon thermometry is calculated as described in L470 and shown in table 6A (reference zircons and 6B (Mount Painter Volcanics).

L662. Attempts to get CA-ID-TIMS ages from Australian S-type igneous rocks often find cores are pervasive enough to make defining an igneous age group impossible, so this is an actual problem in need of a solution.

L668 We are happy to move this paragraph.

L677 We agree that OG1 is the only sample to lose water, which is why we state this in the text. Note that there is an error in the supplementary data table- we have copied the raw data twice instead of also presenting corrected data (which is what we plotted). We apologize for this .

We are happy to check the format of the supplementary tables.

SIMS measurements with low backgrounds and low abundance often have a distribution around zero, tailing in either direction. Replacing all the negative instances will bias the mean, so it is poor practice.

We agree that oxygen mean data should have full uncertainty propagation, but as those are systematic effects they should not be added to the individual spot data in the supplements, but to the aggregate value (in the main text, not the supplement).

From the reviewer's "Scientifically incorrect" section…

L89: The reviewer is wrong. Zero age Pb loss does not change the $^{207}Pb/^{206}Pb$ ratio. The Pb is lost, and because it is modern loss, there is no subsequent ingrowth.

L91, 98: This is disingenuous. Obviously insufficient ion counts will limit precision of any mass spectrometric analysis, SIMS or otherwise. For Pb isotopic ratios, the limit of precision in SIMS is basically set by the precision on the Pb isotopic reference zircon, which is generally between 1 and 2 permil (2 sigma). See Stern et al. (2009) for a more thorough discussion. For U-Pb, there are a host of other issues, such as the calibration, Pb loss, etc., which limit precision more than counting stats for all but the youngest or lowest U zircons. See references in the introduction section for more details, or the introduction of Magee et al. (2023), and references therein, for a more thorough discussion of the calibration and related uncertainties.

L512: The sub-micron positioning of the peizo stage during autoanalysis, particularly in relation to working distance (X in the SHRIMP / American Chemical Society coordinate system), reduces the variability in steering of the secondary ion beam off of the sample (particularly the QT1Y voltage change spot-to-spot).

L513: A 95% confidence interval of 3 Ma out of 1000 Ma (or 1063 Ma ) is roughly three permil, not three percent.

L535: Wee happy to clarify that we refer to the Magee et al. (2016) supplement, not this one.

L554: As we state in L553, the difference in calibration is a function of the difference in slope, and the difference in UO/U.

Fig 7E (actually all Fig 7). As previously mentioned, I copied raw data into the supplement twice, and not the corrected data we plotted. I apologize for this error.

Fig 12. The session plotted is 210046. The original session was 170124. We are happy to edit the caption to make this clear.

References:

As in the manuscript, plus:

Foster, D. A., and Ehlers, K.: (1998), $^{40}Ar$-$^{39}Ar$ thermochronology of the southern Gawler Craton, Australia: Implications for Mesoproterozoic and Neoproterozoic tectonics of East Gondwana and Rodinia, J. Geophys. Res., 103(B5), 10177–10193, doi:10.1029/98JB00151.

Kohn, B. P., Gleadow, A. J. W., Brown, R. W., Gallagher, K., O'Sullivan, P. B. & Foster, D. A.: (2002) Shaping the Australian crust over the last 300 million years: Insights from fission track thermotectonic imaging and denudation studies of key terranes, Australian Journal of Earth Sciences, 49:4, 697-717, DOI: 10.1046/j.1440-0952.2002.00942.x

Vogt, M., Schwarz, W.H., Schmitt, A.K., Schmitt, J., Trieloff, M., Harrison, T.M. and Bell, E.A., 2023. Graphitic inclusions in zircon from early Phanerozoic S-type granite: Implications for the preservation of Hadean biosignatures. Geochimica et Cosmochimica Acta, 349, pp.23-40.

Signed,

Charles Magee, on behalf of the team.

---

## Author Comment (AC2)

Response to reviewer 2

We would like to thank reviewer 2 for their time and effort in reviewing our paper. As mentioned in our response to reviewer 1, we tried to keep figures to a minimum to keep the manuscript size manageable. We apologize if we cut back to the point where our paper became unclear.

With regards to their specific numbered comments:

1. This is an excellent idea, and we would be happy to add a flowchart figure showing sample treatment and analytical pathways.
2. We would also be happy to add concordia diagrams. The table of spot-by-spot analytical results is already available in the supplement table 5, as is the complete reflected light imagery, transmitted light imagery, cathodoluminescence imagery, and spot locations in Supplementary figure 2. As the samples were repolished between U-Pb and $\delta^{18}O$, additional imaging of the U/Pb spots is not possible, but we have post-analytical images from the SHRIMP imaging system saved.
3. We can probably plot ln/ln calibration lines for all fours pairs of reference zircons, but possibly not in the same diagram, due to their very different Pb/U ratios. The reviewer is correct that a calibration slope difference only produces a different age for a different UO/U ratio, which is why we state that difference in line 553.
4. We apologise for mislabeling this diagram, and will fix it. The green line is the median value, and can be removed.
5. We can change the line styles in these figures so that close or overlapping lines can more easily be distinguished.
6. The purpose is to show that the common Pb corrects to a reasonable value using the Stacey & Kramers Pb model. We agree that the trendline of figure 11B is not illustrative of anything and can be removed.
7. This is a good point and we can standardize the format of the concordia plots.

Signed,

Charles Magee, on behalf of the team.

---

## Author Response (AR1)

Both review texts:

REVIEW 1:

Dear Daniela,

I have reviewed the manuscript titled "Effect of chemical abrasion of zircon on SHRIMP U/Pb, δ18O, Trace element, and LA-ICPMS trace element and Lu-Hf isotopic analyses" by Kooymans et al. This manuscript presents a major effort to report U-Pb, O and Lu-Hf isotopic systems and various trace elements including OH of zircon, to show if any effect is observed by chemical abrasion (CA) treatment.

It is indeed a very interesting subject, one that I'd like to see published in the near future, but not in the current form. My biggest concern is that the manuscript could not focus on solving the main problems (which were not clearly defined), but instead distractedly explained too many things and presented data that are not essential. In my opinion, the main argument in this paper (in the other word, the most novel component of this manuscript that zirconologists must be interested in) is that CA could produce poorer surface condition on zircon that possibly affect the sputtering process and change U and Pb ion emissions, resulting in scattered SIMS U-Pb data for moderately damaged zircon.

*The introduction is rewritten to more clearly state what the paper is about.*

The authors did not discuss this well enough but continued discussing other isotopic/elemental data and MPV zircon core data. The other data unchanged by CA are interesting, but unfortunately distracting in the current manuscript, together with the poor writing.

The overall quality of the writing makes the manuscript challenging to navigate. Even the introduction lacks a clear focus on a main purpose, and the motivation and aims of the study are not distinctly described. The manuscript could be significantly shortened by removing repetitive and unnecessary sections.

*Done.*

Many paragraphs consist of only a sentence or two without a clear main idea, indicating a need for substantial improvement in paragraph construction. The overall structural organization is notably lacking, contributing to a distracted and unfocused narrative. A more focused and concise writing style would immensely enhance the manuscript.

Furthermore, many sentences in the manuscript appear as guesses or the authors' opinions without thorough discussion or supporting references or evidence. Oftentimes, conclusions, interpretations, or the significance of these statements are missing entirely. The manuscript lacks a clear logic build-up that is necessary for constructing a reasonable and convincing argument. Throughout my reading, I had to keep asking – "WHY?" or "SO WHAT?". I had to spend much more time than necessary to make a guess what the authors meant. I still do not comprehend many sections and may have misunderstood some parts.

*Agreed, and rewritten.*

I strongly recommend a comprehensive rewrite of the manuscript, with a narrowed focus on SIMS U-Pb data (possibly with O). Considering my suggestion, non-SIMS sections are not much commented below. Expecting a thorough rewriting, I may not provide detailed comments on every aspect I have noted.

Please consider my review anonymous.

**Specific comments**

McKanna et al (2023, GChron) conducted a comprehensive surface study of CA-zircon, and mentioned "sponge-like texture" in CA-treated high-damage zircon samples. They also reported that "acid …regularly accesses crystal cores to dissolve … interior zones", implying that the zircon core also can get the sponge texture if it is highly damaged. That is readily critical to SIMS analysis, which is a very good motivation of this manuscript. McKanna's work is a key paper to introduce the aims of this study.

*Addressed in discussion.*

Scattered U-Pb data observed from CA-treated MPV zircon rims unquestionably raise concerns about the reliability of SIMS U-Pb data for CA-zircon and necessity of CA in any SIMS work. Two hypotheses were introduced to explain it, but they were only briefly addressed in a sentence or two, L603-610, requiring further elaboration and explanation.

*Addressed in discussion rewrite. Obviously further work is required.*

Additional session to re-run the zircon grains where the scattered U-Pb data was a good idea, although would have preferred a more comprehensive double-check, including all the same grains rather than just those deviating from the average U-Pb dates – so that you can properly see if the data are less scattered with >0.05 probability. And the conclusion that sputtering difference on CA-zircon surface (nanospongeform - not visible at all with high mag SE?) is reasonable. But again, I'd like to see more comprehensive discussion about this conclusion.

The argument that CA of moderately-strongly damaged zircon yields poorer SIMS U-Pb data should be consistently maintained throughout the manuscript, not trying to interpret CA inherited zircon data.

*Both the discussion and the introduction have been expanded to address these issues.*

Without considering the possibility of worse SIMS U-Pb data after CA-treatment more, the authors just kept discussing about the MPV core ages in the geological context, which is insignificant because the MPC core age data are highly likely less reliable than MPU. Need discussion if the CA reaching depth into zircon is throughout the core area too (yes, according to McKanna et al 2023). The probability density diagrams of CA-treated and untreated MPV cores – the latter looks a lot sharper, which may be consistent with the possibility of MPV CA cores provided more scattered U-Pb data, same as on the rims. + I'd like to see the diagram of the source rock if you can specify it.

*We have added concordia diagrams for the complete core + rim MPU and MPC data plotted on the same plot for better comparison.*

I also found the higher ratio of the magmatic age (430 Ma) out of total number of core analyses potentially misleading and prone to misinterpretation. 6 vs 18 looks obvious difference, but I believe it may be due to a target selection bias for analysis.

*Target bias would work the other way, as we were looking for inherited cores, and they are easier to see in MPC*

As clearly mentioned and illustrated in the CL figures, CA treated MPV zircons have obvious etch channels. It is highly possible that the authors selected better-looking zircons, avoiding visible cracks, inclusions and possibly etch channels too (even unconsciously). As the authors mentioned, the 430 Ma-cores have the least evident etch channels around, making them more likely chosen for anlaysis compared to those surrounded channels (possibly older). What is the ratio of measured cores that show obvious etch channels? - I could identify max 10 cores analysed in the CL image. Additionally, I cannot understand "survivor bias" (L589) explanation.

Those above are the main comments to the SIMS U-Pb data. Some others follow:

For sample selection: No indication of density or magnitude of damage of zircon? It reads like the authors tempted to use U content as a potential indicator in the section 1.2, but the connection between U content and the level of damage is not explained. In the similar regards, it is not clear why MP volcanic zircon was chosen for this study. Clarifying the rationale behind choosing this specific type of zircon would enhance the reader's understanding of the study's context and objectives.

*The MPV were chosen as the simplest 2 component system (sedimentary core + igneous rim) we had in our collection. This is addressed in the introduction under 2.1 Sample Selection.*

U-Pb internal/external error: you always should report the U-Pb dates with external error propagated, as the published dates and ages are going to be compared to the other ages. Not sure why the authors consider reporting the internal errors only, all the explanations about which make the manuscript at least 1-2 pages longer.

*Internal errors are correct when comparing data from the same session with each other, to look for subtle instrumental differences. External errors are correct when comparing results to reference values. We do both in this paper, which is why we take the time to explain them.*

Untreated TEM2 was used as a primary standard for U-Pb (with two other untreated zircon references) for running both CA-treated or untreated unknowns. This kind of test should be done under the same condition, so it is a reasonable question that any other readers might have. What if the U-Pb calibrations are different between CA-treated and untreated zircon? You showed TEM2 has same U-Pb calib curve slope for both preparations in the results, but it was supposed unknown during the analytical strategy stage (and QGNG and OG1 show different calibration slopes, although the authors consider it trivial; I'd like to see the calibration curves illustrated in the manuscript). If it was deliberate to see how untreated standard affects to CA-treated unknown zircon U-Pb calibration, it should be mentioned.

*We have removed calibration slope discussion at the suggestion of the editor, as it is not the main point. We present data using both T2C (figure 4; table 2b) and T2U (figure 3; table 2a) as the primary reference material, so that the differences can be compared.*

CommonPb correction using 204Pb. L265: "…as 204Pb overcounts were within uncertainty of zero…" …? What else would you do if it is above zero then? And why don't you use 207Pb-corrected 6/38 age for MPV zircon dates?

*Introduction rewrite specifies that we are looking for consistent data treatment for all samples, not the most precise treatment for each sample depending on its age.*

Many unnecessary – examples:

L413: "…did not run as smoothly…" unless you are going to point something out for the less smooth session, it is super unnecessary. Every session has its own condition, and you cannot compare them all the time. L271: Truly unnecessary. Calibration slope could be different even session to session on a same instrument.

*We have removed calibration slope discussion as tangential.*

Many unclear – examples: L24, "…the analyses of chemically abraded materials show excess scatter" OF WHAT?

*Rewritten.*

L107: "These volcanic zircon rims are also lower in U content than S-type granite rims, which often go metamict…" S-type granite "zircon" rim does not have constant U concentration. Which specific S-type granite do you mean? And what is the U content of the MP volcanic zircon? How do you say something is lower than something without showing their data? What is the U concentration threshold to go metamict? Any reference for that?

*We are referring to LFB Silurian igneous rocks- e.g. the most likely intrusive equivalents to the MPV. Rewritten for clarity.*

L149: "We also include four new aliquots for OGC…" Why? Then they are newly measured not from the literature data? And the new data should be properly reported.

*Intro rewritten to explain that this is the missing aliquot level data from Bodorkos et al. 2009 which is finally being provided.*

Result-discussion mixed up: L287-9, that is one of the aim of this study, not method

L502: "… reduces the uncertainty in the reference ages by 140-290%..." According to Table 1, ratio uncertainty except 91U (why is that?) is certainly smaller including tracer, but not age uncertainties. How come? Age uncertainties are actually larger including tracer.

*Table 1 was misprinted. Fixed.*

L541: "…, which is consistent with our SHRIMP U-Pb data" How and what exactly?

*Rewritten for clarity.*

L549-551: Not sure.. Fig 2 shows QNGN and OG1 have different calibration slopes between U-C

*Removed as tangential.*

L552-554: self-calibrated? What about the other standards data using the low-sloped 91500 calibration?

*Removed as tangential.*

L556-560: that is a pure guess. If you want to say so, you need more examples of low Hf & REE zircon data. Do not argue anything without supporting evidence.

*Removed as tangential.*

L572-578: Why S-type granite only? Slow(er) crystallization is a general condition to form granite than volcanic rock. And common Pb is not only from Pb-rich inclusion. "For comparison…": comparison of what and what? Why the granites from Bodorkos et al (2015)? Apparently same

igneous ages (~430 Ma), so are they potential co-genetic bodies? There is no explanation. "…statistically significant common Pb": which means…? "…raw 207Pb/206Pb ratios…": raw ratios? Same as total 207/206? 207, 206Pb are mostly radiogenic Pb in zircon anyway? "…unusually high common Pb contents" I think it is not uncommon to see common Pb? Apart from many question marks about every single sentence in this paragraph, I do not see why this full paragraph is necessary.

*Rewritten for clarity*

L582-3: show the data. Compare the size of uncertainties or etc.

*Figures added to paper.*

L608-9: Uh… probably MPV zircon is more damaged than the reference zircons?

*Could be.*

L613: total 206Pb… why?

*To constrain potential diffusion gradients (which were not observed).*

L617-: I don't get what you try to explain in this paragraph.

*Removed.*

L627-633: "This is consistent with…" "consistent" is to compare to the others' arguments or consensus. To me, this paragraph is a guess, with no other supporting arguments. From the rim data (MPU, MPC), I am convinced that the CA method weakens the SIMS U-Pb validity and think the MPC core dating is less reliable due to the CA-induced surface damage (nanospongeform – according to the authors' description, L607).

*We have rewritten much of the U-Pb introduction and discussion to explain why this is generally not the case.*

L650-: That is bizarre to publish. I strongly suggest the authors to re-run a session for Temora2 using a same batch. What if the huge d18O difference after CA is true?

*We would love to as well, but we will not be able to do this until at least 2025 due to scheduling commitments, and this work is already 7 years old…*

L654: MPV zircon scattered d18O may indicate that the source rock/melt was not in equilibrium. Not all S-type granites show scattered zircon d18O!

*Rewritten to compare to Silurian Lachlan Fold Belt samples, not worldwide S-type granites (See Vogt et al. (2023) for chemically abraded Variscian S-types).*

L657-8: what kind of evidence do you need?

*Rim d18O does not seem to correlate with core d18O.*

L658: Ti thermometry? Reference? Did you perform the measurement?

*Reference is given in the table caption and the methods, and measurement results are in the tables on a sample mean bassi, and the supplementary data on a spot-by-spot basis.*

L662: "… not all … volcanic zircon cores are detrital" They never are.

*Within the Silurian LFB they are nigh ubiquitous.*

L668-: this paragraph should to go the result section

*Fair.*

L677-8: I don't see significant difference in Fig 7 except OG1, which look plotted wrong

*Supplementary data table and methods corrected to explain.*

**Technical corrections**

*Supplemantary tables:*

Check all the table format (not only for the suppl tables)

*For what?*

All negative values should be corrected to zero or "-"

*We disagree, due to issues arising from biasing averages to positive values.*

Th/U not a ratio of isotopes (232Th/238U) but total Th/U estimate

Reasonable effective number and decimal places

Oxygen data table only show internal SE (standard error), which should be replaced by fully propagated uncertainties. (or if you argue that no external reproducibility is not necessarily considered because only within-session data are compared, it should be explained in the text)

*Figures:*

Fig 1: it is useful to have some indications to figure our easily which cores are syn-eruptic and which "cracks" are the etch channels.

*Full annotation is in the supplementary figures.*

Fig 4: readers would like to see the Concordia plots for all standard/samples.

*Done.*

Fig 11: CA treated zircon data should be included too.

*Done.*

*Other comments*

It does not read a formal scientific manuscript, rather oral speech script or a personal journal in many aspects – use of informal words (best example is SHRIMPing), use of subject adjectives without giving numbers, lots of mistakes, sentences scientifically incorrect…

Lots of mistakes: typos, wrong capital letters, use of spaces, inconsistent expression (U-Pb vs U/Pb, use of acronyms), etc

Always make it clear what uncertainty you indicate – 1se, 1sd, 1s (65% conf) or 2s/ts (95% conf) etc.

*Checked.*

SHRIMP – a specific brand name, but a method. Change it to SIMS except where specific SHRIMP IIe and SI are described.

*Checked.*

*Scientifically incorrect or text-data-plot not matching:*

L89, 207/206Pb ratio is almost constant: not necessarily especially for old ones (in that regards, I actually would like to see Pb/Pb data too especially for OG1)

*Shown in new plots.*

L91: "… SHRIMP can produce 7/6 age with 2‰ precision (really meant permil not percent?)" It cannot be a general comment as 7/6 age precision depends on the zircon age and Pb concentration.

*Checked.*

L98: "…old enough for decent counting statistics…" Again, not necessarily. Counting statistics of U-Pb dating depends on U concentration too.

*Sure, but high U zircons come with their own problems, which are beyond the scope of this study. Try Magee et al. (2017) for discussion of the hi-U effect..*

L512: "a piezo stage, automated analyses" why should they be conditions to get better precision?

*Better X axis reproducibility during automated analysis.*

L513: "…better than the 1-3% value…" Better? you see the exactly that range from TEM2 and 91500 in Table 2 (and larger for the other standards)?

*Table two shows an external 95% confidence interval of 1.35 to 1.8 Ma, out of 417 Ma = 0.43%. This is less than 1%. It shows an external 95% confidence envelope for 91500 of 3.6-5 Ma, out of 1063 Ma = 0.47%. These numbers are less than one percent.*

L535: you do not have DR12 and 13 figures; if they are the ones from Magee et al., those fig numbers are not necessary or you need to make it clear.

*Clarified.*

L554: "For a difference in slope of 0.5, this would yield ages 0.2% older…" Calibration slope does not necessarily increase or decrease ages, as it is depending on where the unknown data sit compared to the standard's calib curve.

*Removed for clarity.*

Fig 7E: OH/18O values of OGC are all >0.1 in the suppl table, but they are plotted mostly <0.1 in thei figure.

*Supplement fixed to show both raw and corrected data, and methods updated to explain correction.*

Fig 12: The repeated session is 210046? It is 170124 session in the fig

*Clarified.*

REVIEW 2

This manuscript is dealing with the topic which all geochronologist would like to know the answer. Careful design of whole experiment as well as precise analysis using various instrument (TIMS, SHRIMP, LA-ICP) should have a huge contribution and implication on the area of geochronology and geology. This result could be cited many times in future and be possibly mentioned in the textbook, too.

However, the manuscript itself is not very straightforward for reader to follow and understand. Most materials of tables and figures are not ready to be published yet. Significant revision in both text, table, and figures are necessary in text and table and figures for the next step. (I agree many of things which referee #1 pointed out and will not repeat that here. )

1. Since authors are trying many experiment, especially for comparisons 1) between untreated and chemically abrasion and 2) among four well-known reference zircons and more, the summarized graphic including procedure and results would be effective way to make clear the output of this research.

*We tried making such a graphic, but it wasn't very clear, so we didn't submit it.*

2. For zircon from Mount Painter Volcanics, cathodoluminescence imaging (figure 1) and probability density diagram (figure 5) are not enough. Additional concordia diagram, table for all values, and CL imaging with a higher magnification including the beam spot after SHRIMP analysis will be necessary for the argument.

*Additional concordia diagrams, with consistent format, are provided for all SHRIMP (and some TIMS) U-Pb data.*

3. The slope of SHRIMP calibration in the diagram of ln(Pb/U) vs. ln(UO/U): it will be better to present all dataset on (Pb/U) vs. ln(UO/U daigram with figure 2. The argument in the line 554-555 is not valid because the slope itself cannot change the date of each spot analysis. The combination of slope AND UO/U value of each spot can be affected the calibrated date.

*This part of the manuscript is a distraction, so we removed that part of the manuscript.*

4. Figure 3: last diagram should be T2C not 91U. and what does green line in the middle means?

*Fixed. The green line in the central value as output by Isoplot.*

5. Figure 6 and figure 10: dashed line (green and blue) is not well recognized.

*We are not sure how to fix this while maintaining a colourblind-friendly palette.*

6. Figure 11: what is the reason to show both A and B? Moreover, the upper intercept of age of figure 11-B seems to be meanlingless.

*We have replotted this data in a format consistent with the other plots.*

7. Figure 4, 11, and 12: No consistent format of all concordia plots. Especially Figure 12 are too confused diagram and it is very hard to get the point.

*Fixed.*

---

## Author Response (AR2)

**Dear Editor,**

Everything crossed out in the copy of your notes appended to this note has been addressed. As for the other stuff:

It is important to at least mention Lu-Hf, as that was an important driver for the study and a justification for why we were allowed to do this project. As that portion of the project ended up being fairly uninteresting, we have reduced the content, but we feel it is important to let the community we checked this- these days most zircons dated by SIMS end up going for Lu-Hf eventually, and whether or not CA disturbs in-situ Lu-Hf analysis is important to know for people deciding whether or not to CA their samples.

The lines relating to upgrades to the SHRIMP (245-248) are useful in case other labs try to reproduce these results. We find that the new stage gives much lower variation in QT1Y steering- e.g. more consistent ion trajectories from the sample surface to the mass spectrometer. We don't know if this is necessary for achieving highly accurate and precise SIMS U-Pb results (e.g. better than 0.5%), but we feel it is worth mentioning so that other labs who try to reproduce these results know exactly what setup we are using.

Crossout below has been addressed:

Title, abstract and throughout: The data you present are relevant to all SIMS, not just SHRIMP. Please replace this term (; I know they are well known, but geochronology aims to reach a wider audience.

Title: correct the inconsistent use of capital letters (), and I would remove Lu-Hf as this is really a minor aspect.

Abstract: , then report data and results.

Throughout: I recommend using , as for Lu-Hf or Rb-Sr, where Pb/U is the ratio measured.

 to 3440 and 3465 Ma you mention.

245-248, this upgrade to the SHRIMP is not really relevant for this study.

Line 440-448, merge paragraphs.

Line 468, 487, Table should be capital, check throughout.

Line 473, "All results are μg/g" this information should be in the Table notes, not in the main text.

Section 3.7.1: describe results in the present form.

Line 491, do not introduce the BLD abbreviation as it is not further used in the text.

Line 494-95: this belongs to Methods.

Line 506: correct to "reason we think"

Line 508: change to "consideration of"

Line 509: specify the method used in Magee et al. 2023.

Line 550-553, please explain better what you men here: how do you know the exact age of the plunton if zircon is dated at 3440 Ma?.

Line 553-554, join paragraphs.

Line 573, rephrase "the common Pb composition is close to that predicted by the model (ADD REREFERENCE)". Also consistently use Pb instead of lead.

Line 584-615: merge paragraphs where meaningful, a number of them seems isolate thoughts.

Line 625: change to "Fig. 1".

Line 647: give uncertainty of the Ti-in-zircon T estimate (at least a 2 SD, or a fully propagated uncertainty).

Line 686: "only the initial Hf composition of the 91U was not within uncertainty of the reference values" this may deserve a comment, is 91500 not a reliable Hf isotope standard?

703: what do you mean "when the SHRIMP is running well"? Would data be published of bad runs? And what is the definition of "well"? You may want to rephrase this to "SHRIMP……. can achieved accuracy and precisions of..." and for which type of data?

Thank you for redrawing some of the figures. They are all of high quality, just avoid yellow lettering on a white background, Fig. 13.

Please combine Figs 9 and 10 into a single figure.

Figure 11 could be better paginated: 4 graphs of the same size, properly aligned and possibly without outer margins. Horizontal axis lines could be lighter or dotted so as not to crowd the plots.

Tables 6, 7 and 8 should be moved to the supplement.

I look forward to seeing the revised version of the MS.

Kind regards
Daniela Rubatto
Associate Editor